# Toward hyper-resolution global hydrological models including human activities: application to Kyushu Island, Japan

Naota Hanasaki[1], Hikari Matsuda[1], Masashi Fujiwara[1], Yukiko Hirabayashi[2], Shinta Seto[3], Shinjiro Kanae[4], Taikan Oki[5]

[1]National Institute for Environmental Studies, Tsukuba, Japan
[2]Shibaura Institute of Technology, Tokyo, Japan
[3]Nagasaki University, Nagasaki, Japan
[4]Tokyo Institute of Technology, Tokyo, Japan
[5]University of Tokyo, Tokyo, Japan

*Correspondence to*: Naota Hanasaki (hanasaki@nies.go.jp)

**Abstract.** Global hydrological models that include human activities are powerful tools for assessing water availability and use at global and continental scales. Such models are typically applied at a spatial resolution of 30 arcminutes (approximately 50 km). In recent years, some 5-arcminute (9-km) applications have been reported, but with numerous technical challenges, including the validation of calculations for more than a million grid cells and the conversion of simulation results into meaningful information relevant to water resource management. Here, the H08 global water resources model was applied in two ways to Kyushu Island in Japan at resolution of 1 arcminute (2 km), and the detailed results were compared. One method involved feeding interpolated global meteorological and geographic data into the default global model (GLB; in accordance with previous high-resolution applications). For the other method, locally derived boundary conditions were input to the localized model (LOC; this method can be easily extended and applied to other regions, at least across Japan). The results showed that GLB cannot easily reproduce the historical record, especially for variables related to human activities (e.g., dam operation and water withdrawal). LOC is capable of estimating natural and human water balance components at daily time scales and providing reliable information for regional water resource assessment. The results highlight the importance of improving data preparation and modeling methods to represent water management and use in hyper-resolution global hydrology simulations.

## 1 Introduction

Global hydrological models have successfully incorporated major natural hydrological components as well as a number of important human activities, including agricultural water use (Döll and Siebert, 2002;Wada et al., 2013), industrial and

domestic water use (Alcamo et al., 2003a;Flörke et al., 2013), reservoir operation (Hanasaki et al., 2006;Shin et al., 2019),

groundwater pumping (Wada et al., 2010), desalination (Hanasaki et al., 2016), and aqueduct water transfer (Hanasaki et al., 2018b). Several global hydrological models are capable of simulating global water availability and use, including most of these human activities, for the purpose of water management. Such models include WaterGAP2 (Alcamo et al., 2003a;Müller Schmied et al., 2020); LPJmL (Rost et al., 2008;Schaphoff et al., 2018); H08 (Hanasaki et al., 2008a;Hanasaki et al., 2018b); WBMplus (Wisser et al., 2010); PCR-GLOBWB (Wada et al., 2011;Sutanudjaja et al., 2018); HiGWMAT

(Pokhrel et al., 2012;Shin et al., 2019); VIC5 (Droppers et al., 2020), and CWatM (Burek et al., 2020). These models have been widely employed for water scarcity assessment (Alcamo et al., 2003b;Hanasaki et al., 2018a), climate change impact assessment (Schewe et al., 2014), water footprinting (Hanasaki et al., 2010;Dalin et al., 2017), estimation of Earth's carrying capacity (Gerten et al., 2020), and many other purposes.

Among various challenges facing global hydrological models, increasing spatial resolution is perhaps the most

straightforward and most difficult. For the past decade, most global hydrological simulations have been conducted at a spatial resolution of 30 arcminutes (0.5°, approximately 55 km at the equator). A widely recognized goal is to realize a spatial resolution of 30 arcseconds (approximately 1 km at the equator), designated hyper-resolution global land surface modeling, which has been described by Oki et al. (2006) and Wood et al. (2011). With increased spatial resolution, human activities become non-negligible, as they predominate in highly developed and densely populated regions. Establishment of

methodologies to incorporate human activities at such a scale is urgently needed.

Recently, some models have been applied at spatial resolutions considerably finer than 30 arcminutes. To the authors' knowledge, the first such example was reported by Eisner and Flörke (2015). They developed the WaterGAP3 model based on WaterGAP2 (Alcamo et al., 2003a) and applied it globally at 5-arcminute resolution. Wada et al. (2016) applied the PCR-GLOBWB model globally at 6-arcminute spatial resolution. They estimated sector-wise water usage along with the supply

of surface and groundwater, and then calculated a conventional water stress index, the ratio of usage to supply. They found that water stress is unrealistically concentrated in grid cells containing the downtown areas of the largest cities (e.g., Paris, New York) and therefore recommended assessment across larger spatial domains, such as sub-basins and counties. Sutanujaja et al. (2018) developed PCR-GLOBWB2 as an upgrade of PCR-GLOBWB, which has two standard spatial resolutions, 30 and 5 arcminutes. They reported that the latter resolution outperformed the former, mainly due to better

representation of snow processes by its more detailed air temperature distribution in mountainous areas. Burek et al. (2020) developed a new global hydrological model called CWatM based on PCR-GLOBWB and LISFLOOD (De Roo et al., 2000). Although no detailed results were reported, similar to PCR-GLOBWB2, boundary conditions for 5-arcminut resolution have been determined for this model.

Although several successful applications have been described, global hydrological modeling at scales finer than 30

arcminutes is still in its infancy. Such modeling efforts face three major challenges. The first challenge is validation. To demonstrate that the models are capable of operating at a certain spatial resolution, reliable observations must be collected in advance. However, aside from river discharge, reliable global observations of hydrological variables are challenging to

obtain, especially fine-scale data for water use and management. The second challenge is model refinement. To express site-specific hydrological processes and water use practices, parameter regionalization (Beck et al., 2016;Yoshida et al., 2021) and model and data localization processes are required, and these processes can easily become unmanageable unless conducted systematically and strategically. The third challenge is inter-grid-cell connection. Because water supply and drainage systems for urban and irrigation water distribution typically cover a greater extent than the 5-arcminute grid cell, inter-connections must be considered.

To investigate the applicability of a global hydrological model at fine spatial resolution, we applied the latest version of the H08 global hydrological model (Hanasaki et al., 2018b) to Kyushu Island, Japan, at 1-arcminute spatial resolution (approximately 2 km at the equator). The study area provides sufficient quantity and quality of data for simulation and validation with reasonable efforts. The site includes major rivers, dams, irrigation projects, active industries, and densely populated cities, providing good example locations for application of the hyper-resolution global hydrological model. Similar efforts have been reported by Mateo et al. (2014) and Masood et al. (2015), who applied H08 at 5-arcminute spatial resolution to the Chao Phraya River in Thailand and the Ganges-Brahmaputra-Meghna Rivers of the Indian Subcontinent, respectively. The present study uses the latest H08 model at five-fold higher resolution than previous works, with a focus on water management and use. Throughout this study, we assess water resources in terms of whether water is available when and where it is needed. Other assessments, such as flood risk, are outside the scope of this research. The present study addresses the following three questions. How well can the default global H08 reproduce the hydrology and water management of Kyushu Island? Does localization of the data and model improve the performance? What is the benefit of conducting hyper-resolution hydrological modeling from the viewpoint of water resources management and assessment?

## 2 Study area

Kyushu is the third largest island in Japan, with an area of 36,750 km$^2$. Kyushu is surrounded by the Japan Sea to the north, the East China Sea to the west, and the Pacific Ocean to the south. The terrain is mountainous, and flat areas are mostly located along the downstream reaches of major rivers. All of Kyushu experiences a humid subtropical climate (Cfa in the Köppen-Geiger climatic classification). Mean annual precipitation is approximately 1600–2500 mm year$^{-1}$, and generally increases southward. The precipitation is concentrated from June to July, when a seasonal rain front crosses the island. Additionally, typhoons occasionally bring considerable rainfall from June to October. It snows only occasionally. Because of its mountainous topography, the catchment areas of rivers on Kyushu are relatively small. The largest river basin is the Chikugo River in north-central Kyushu, which is 143 km in length with a basin 2860 km$^2$ in area.

In this study, we targeted nine rivers with catchment areas exceeding 1000 km$^2$ (Table 1). A total of 420 reservoirs are listed in the dam handbook issued by the Japan Dam Foundation (http://damnet.or.jp/Dambinran/binran/TopIndex_en.html). Due to the steep terrain of Kyushu, the storage capacity of dams on the island is relatively small. We primarily targeted 13 existing dams with storage capacity exceeding $40 \times 10^6$ m$^3$. Among these dams, eight are located in the nine major river

basins included in this study (Table 2). Kyushu has seven prefectures, with a total population of $12,930 \times 10^3$ in 2015 (https://www.e-stat.go.jp/en). Total cropland area is 5500 km$^2$. The dominant crop types are paddy rice and a variety of vegetables and fruits. Total water withdrawal on the island is 9.8 km$^3$ year$^{-1}$, of which 7.3 km$^3$ year$^{-1}$ is used for irrigation (Ministry of Land Infrastructure Transportation and Tourism (MLIT), 2021).

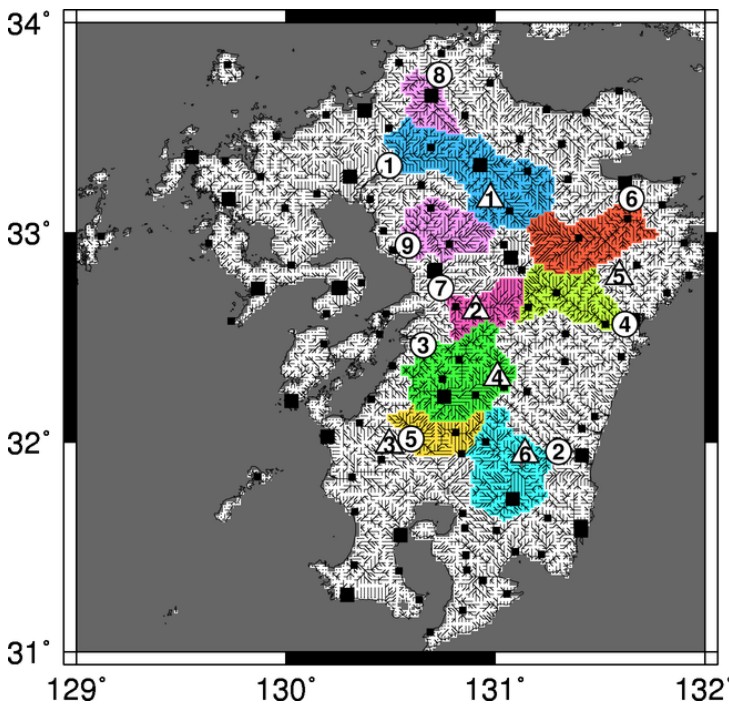


**Figure 1 Study domain. Large and small squares represent manned and automated weather stations, respectively. Numbers in circles indicate gauging stations along major rivers. See Table 1 for a list of rivers and gauging stations. Colored areas are the catchment areas of the gauging stations. Numbers in triangles indicate dams located on the major rivers. See Table 2 for a list of dams.**


**Table 1: Major rivers of Kyushu Island. The gauging stations are the official reference points of rivers where a long-term streamflow record is obtained for planning of management and development, with two exceptions. The official reference point for the Chikugo River is Arase (1443 km$^2$) but, for wider coverage of the catchment, we used the downstream station Senoshita. Similarly, the Sendai River is recorded at Sendai (1425 km$^2$), but because that site is affected by the operation of the Tsuruda Dam,**
**we used the upstream station Suzunose.**

| ID | River | Catchment area [km$^2$] | Gauging station | Catchment area [km$^2$] |
|----|-------|------------------------|-----------------|------------------------|
| 1 | Chikugo | 2860 | Senoshita | 2295 |
| 2 | Oyodo | 2230 | Takaoka | 1564 |

| 3 | Kuma | 1880 | Yokoishi | 1856 |
| 4 | Gokase | 1820 | Miwa | 1044 |
| 5 | Sendai | 1600 | Suzunose | 720 |
| 6 | Oono | 1465 | Shiratakibashi | 1381 |
| 7 | Midori | 1100 | Jonan | 681 |
| 8 | Onga | 1026 | Hinodebashi | 695 |
| 9 | Kikuchi | 996 | Tamana | 906 |

**Table 2 Major dams along the rivers included in this study. We excluded the Ryumon Dam due to a lack of historical observation data. We excluded the Matsubara Dam because it is located just below the Shimouke Dam, complicating its use for validation.**

| ID | Name | River | Capacity [$10^6$ m$^3$] |
| --- | --- | --- | --- |
| 1 | Shimouke | Chikugo | 59.3 |
| 2 | Midorikawa | Midorikawa | 46.0 |
| 3 | Tsuruda | Sendai | 123.0 |
| 4 | Ichifusa | Kuma | 40.2 |
| 5 | Kitagawa | Gokase | 41.0 |
| 6 | Iwase | Oyodo | 57.0 |
| - | Ryumon* | Kikuchi | 42.5 |
| - | Matsubara** | Chikugo | 54.6 |

## 3 Methods

In this study, we applied the global hydrological model H08 (Hanasaki et al., 2018b) to Kyushu Island in Japan at a spatial resolution of 1 arcminute. We ran the H08 in two ways: using the standard global model, and using a modified model. H08 requires two types of boundary conditions, namely meteorological information at daily intervals and static geographic information. Boundary conditions were also prepared in two ways. The first set of conditions was obtained using global data. We attempted to find data at 1-arcminute or finer resolution, but most data were available only at 5-arcminute or coarser 120 resolution, which we linearly interpolated into 1-arcminute data. The second method for obtaining boundary conditions employs local data from Japanese governmental agencies and other sources, which are compiled into 1-arcminute resolution. The combination of the default model with global boundary conditions is the current typical implementation of global hyper-resolution simulations, while the localized model with local boundary conditions represent an ideal implementation when

data accessibility issues have been resolved (i.e., assuming hydrometeorological data are available at high resolution for any location worldwide).

## 3.1 Model

### 3.1.1 The H08 model

H08 was originally developed to simulate the global hydrological cycle with major human activities for advanced water resource assessment at the global scale (Hanasaki et al., 2008a;Hanasaki et al., 2018b). The model is grid-based and its standard spatial resolution is 30 arcminutes, but it was designed for application to any spatial domain and resolution. The model consists of six sub-models, namely land surface hydrology, river routing, crop growth, water abstraction, reservoir operation, and environmental flow. These components enable simulation of water availability and use at daily intervals in a fully consistent manner and division of the water used by three sectors (agricultural, industrial and municipal) into five surface water sources (river, aqueduct, local reservoir, seawater desalination, and unspecified surface water, which is defined below) and two groundwater sources (renewable and non-renewable groundwater).

The crop growth and water abstraction sub-models are detailed below, as they play particularly important roles in this study. The crop growth sub-model estimates the daily growth of crops using the heat-unit approach and biomass accumulation based on photosynthesis. The formulation of this sub-model is identical to the widely used Soil and Water Assessment Tool (SWAT; Arnold et al. (2012)). The crop growth sub-model has two functions in H08. First, it is a component of H08 that outputs crop-related variables fully consistent with global water cycle simulation, such as crop yield. Second, it is a stand-alone model for estimating the crop calendar in the study area. Crop calendars, specifically planting and harvesting dates, are essential for determining the timing of irrigation, but local information is difficult to obtain at any scale, as these dates are rarely monitored and recorded. The principal purposes of the stand-alone simulation were to iterate the crop yield estimation 365 times by shifting the planting date from January $1^{st}$ to December $31^{st}$ in all grid cells and, thereby, to identify the date that results in the maximum crop yield.

The water abstraction sub-model also plays an essential role in H08. The fundamental concept used for modeling of each water source is explained below. Water abstraction from rivers indicates the removal of a portion of streamflow to meet the water requirement within a grid cell. Streamflow is estimated by accumulating runoff through the river network. Water abstraction from an aqueduct indicates transfer of streamflow beyond a grid cell through the manmade aqueduct network. This network is defined either explicitly (i.e., manual connections are made between source and destination grid cells) or implicitly (i.e., the origin and destination of aqueducts is inferred by assuming gravitational water transfer via an algorithm, see Hanasaki et al. (2018b) for details). Water abstraction from a local reservoir indicates abstraction of water stored in relatively small reservoirs or irrigation ponds within a grid cell that do not affect streamflow of the river network. Note that large reservoirs are incorporated into the river network and directly regulate streamflow. Water abstraction from seawater

desalination indicates abstraction of water processed in desalination plants, which is possible only in designated regions. Water abstraction from unspecified surface water indicates the volume of the daily water requirement that was from the sources listed above; it indicates that the water requirement was not met (i.e., water deficit) or that errors occurred in simulation (e.g., underestimation of runoff, overestimation of demand, or misrepresentation of water infrastructure). Water abstraction from renewable groundwater indicates water abstraction from the groundwater reservoir of the cell, which can be recharged. Abstraction from non-renewable groundwater indicates deep groundwater pumping after depletion of the groundwater reservoir. For simplicity, we excluded local reservoirs and seawater desalination from this study. Thus, the water requirement is divided into river, aqueduct, unspecified surface water, renewable groundwater, and non-renewable groundwater.

### 3.1.2 Land surface hydrology sub-model localization

To reflect basin characteristics more accurately, we localized some sub-models of the original H08 model. For the land surface sub-model, we tuned the hydrological parameters. Numerous methods have been proposed for tuning of parameters in hydrological models, and we adopted a simple Monte-Carlo based method described previously (Mateo et al., 2014;Masood et al., 2015). This process identifies the combination of four sensitive hydrological parameters that minimizes the Nash-Sutcliffe efficiency (NSE; (Nash and Sutcliffe, 1970) of the daily streamflow simulation among three possible values for each parameter (total $3^4 = 81$ parameter sets), namely the default, upper, and lower values (see Supplementary Text and Table 1). We applied this method at nine major river gauging stations (Table 1). This approach is applicable to gauged basins, but not ungauged basins. To tackle this problem, we identified the most frequently selected value for each parameter as the best option for application across all of Kyushu Island. Then, we used the selected parameter values throughout the period and in all grid cells.

### 3.1.3 Reservoir operation sub-model localization

The reservoir operation sub-model incorporated into H08 employs simple reservoir operation rules to determine daily water release and storage for individual reservoirs from limited globally available information (Hanasaki et al., 2006). The validity of the original algorithm as a generic global model has been demonstrated in numerous previous works (Masaki et al., 2017;Yassin et al., 2019;Shin et al., 2019;Turner et al., 2020). The original reservoir operation sub-model (see Hanasaki et al. (2006)) releases water at a constant rate throughout the year at a rate equivalent to the mean annual inflow. This process results in increased storage during the wet season and reduced storage in the dry season. When no other control is applied for sub-annual operation, storage can reach the storage capacity of the reservoir (and then begin to overflow) or become completely depleted (so that release becomes identical to inflow), which seldom occur in reality (Masaki et al., 2017).

To enable more realistic behavior, we added the release curve and the upper and lower storage curves to the model (Figure 2). The release curve assigns different release rates for the wet and dry seasons. The upper and lower storage curves determine

the maximum and minimum storage in a year. The storage curves are subdivided into four phases, namely maintaining high storage, releasing, maintaining low storage, and recharging. In the releasing phase, the maximum and minimum storage values are gradually decreased to avoid a sudden increase in water release. In the recharging phase, the maximum and minimum storage values are increased to allow fast and effective recovery. The release curve is also subdivided into four phases, namely the dry season, dry to wet transition, the wet season, and wet to dry transition. To represent the various shapes of release and storage curves, we constructed a generic curve, as shown in Figure 2. To represent these curves, we specified four points (R1–4, in time and release coordinates) on the release curve, and a total of eight points (U1–4 and L1–4, in time and storage coordinates) on the upper and lower curves. R1–4 were determined from the long-term mean monthly release levels of individual reservoirs. U1–4 and L1–4 were determined from the actual operation curves compiled by reservoir operators. Storage curves of reservoirs managed by central and local governments and public agencies are disclosed due to legislation in Japan. The parameter values used for six dams are listed in Tables S3–5.

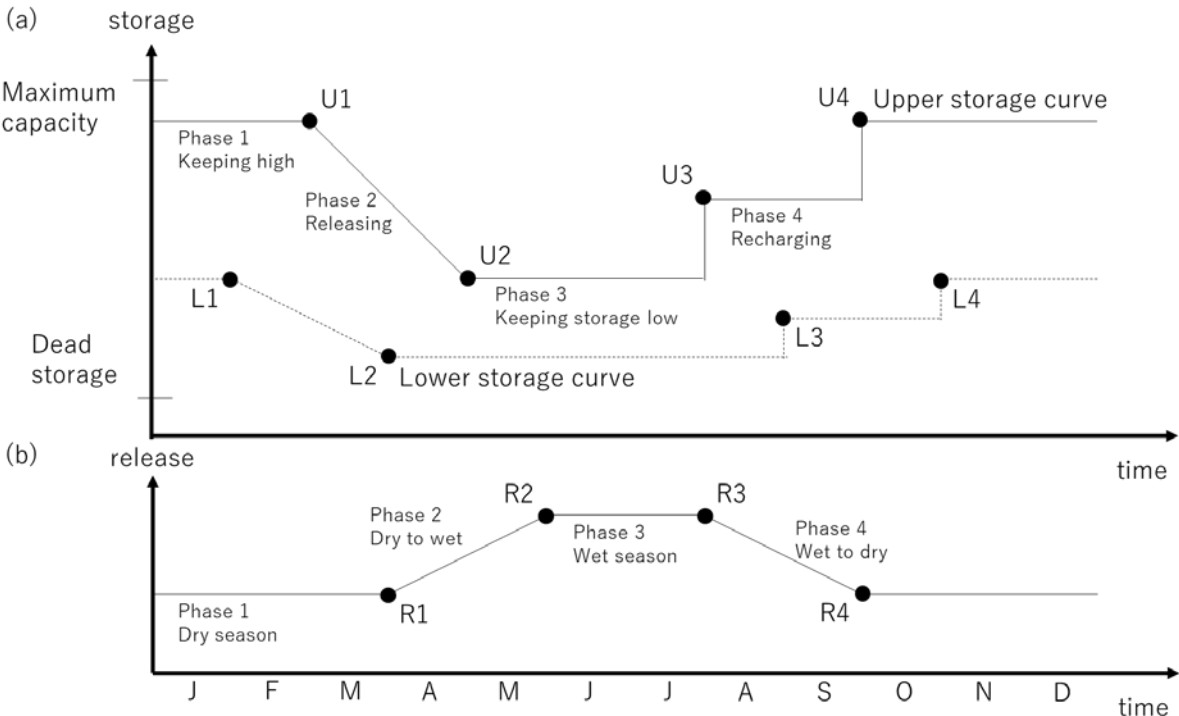

**Figure 2 Schematic diagrams of (a) the upper and lower storage curves, and (b) the release curve.**

200

### 3.1.4 Implicit aqueduct estimation

In reality, to abstract streamflow for water use, water is withdrawn at weirs and transferred directly to the point of use (e.g., irrigated cropland) or to water treatment facilities and then supplied to urban areas through pipelines. Because identification of weirs and manual delineation of water transfer routes is infeasible for large study domains, H08 can infer the route of inter-grid-cell gravity-based water transfer (Hanasaki et al., 2018b). The original algorithm was designed to work at a resolution of 30 arcminutes and, therefore, we modified it for better performance at fine spatial resolution.

We modified two components of this algorithm. First, the original method allows transfer of only 1 grid cell ahead, but was modified to allow transfers a maximum of 5 grid cells ahead (approximately 10 km). Next, inter-basin transfer was not possible in the original model, but was allowed in this study if the catchment area of the destination is smaller than two grid cells. This modification adds flexibility for water transfer, especially in grid cells near coastlines, where basins are generally small (Figure 3). Along with these implicit aqueducts, the origins and destinations of known major water transfer facilities (designated explicit aqueducts, see section 3.2.) were prepared and implemented into the river network (Figure S1; Table S6). These explicit aqueducts transfer water irrelevant to elevation.

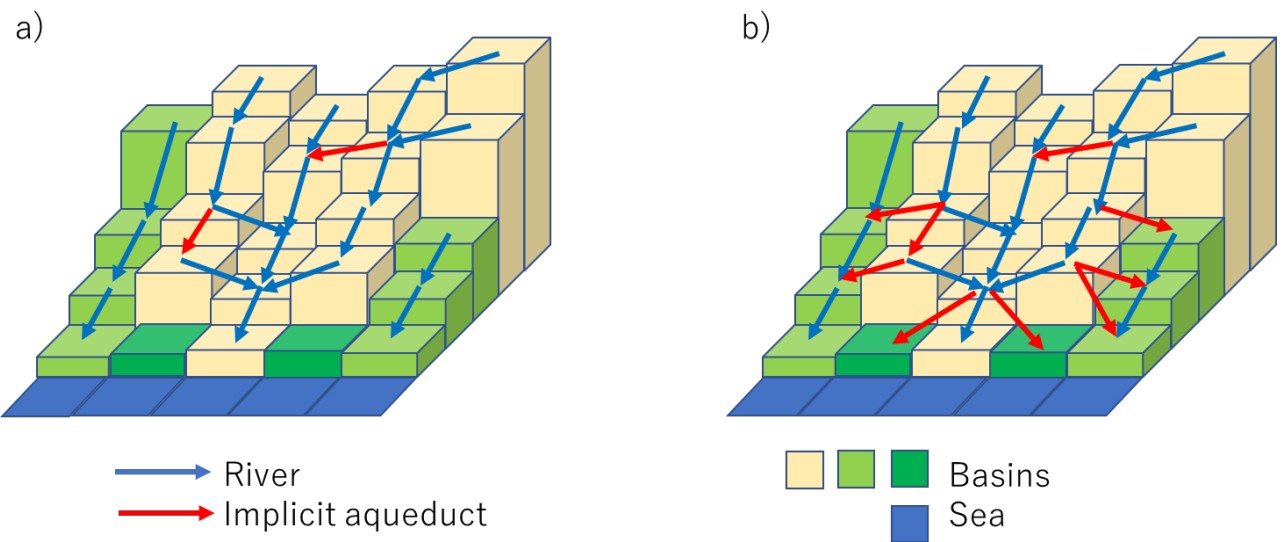

**Figure 3 Implicit aqueducts. a) The original scheme, which does not allow inter-basin transfer. Many coastal grid cells (shown in dark green) are prone to water scarcity because they have limited catchment area. b) The new scheme, which conditionally allows inter-basin transfer. Coastal grid cells receive inter basin water transfer from neighboring basins. Columns represent land grid cells. The height indicates each grid cells' mean elevation. Colors indicate different basins.**

## 3.2 Data

### 3.2.1 Meteorological data

To run the H08 model, seven meteorological variables must be supplied at daily intervals: precipitation, air temperature, humidity, wind speed, surface pressure, precipitation, and downward longwave and shortwave radiation. We prepared the data using two methods, including one based on global data representing the current practice in high-resolution global modeling and one using local data.

For global data, we extracted and linearly interpolated the WFDEI global meteorological dataset (Weedon et al., 2014) for the study domain. The dataset contains all seven variables at daily intervals from 1979–2016. WFDEI is a hybrid product of monthly observation-based global gridded meteorological data, for example, CRU TS3.1 (Harris et al., 2014) and GPCC (Schneider et al., 2014) and the ERA-Interim daily global reanalysis data (Dee et al., 2011).

For local data, we used published meteorological records. Kyushu Island contains 44 observatories (District and Local Meteorological Observatory) and 121 automatic weather stations (Automated Meteorological Data Acquisition System; AMeDAS; https://www.jma.go.jp/jp/amedas/) operated by the Japan Meteorological Agency (Figure 1). AMeDAS monitors air temperature, wind speed, precipitation, and sunshine duration. Sunshine duration was converted into downward shortwave radiation using the method described in Appendix A. The observatories also monitor surface pressure and specific humidity. A total of 117 observatories and AMeDAS stations that provide continuous records were used for this study. The daily observations at observatories and stations were interpolated using inverse distance weighting and converted into grid data. Finally, downward longwave radiation was estimated using the method described in Appendix B.

### 3.2.2 Geographic data – Hydrology

For H08, the boundary conditions listed in Table 3 must be provided. We first prepared all of these parameters by extracting and interpolating their values from the default global simulation or using the functions of H08, which were described in detail by Hanasaki et al. (2018b). Then, we prepared another set of data utilizing local information whenever possible, as detailed below.

We first developed a flow direction map of Kyushu Island. We used the digital elevation model SRTM30 PLUS (Becker et al., 2009). Using geographic information systems (ESRI ArcGIS and ArcHydro), we constructed a flow direction map at 1-arcminute resolution. Then, we georeferenced the six largest dams, which are listed in Table 2, to the flow direction map. Next, we prepared an implicit aqueduct network with the localized algorithm (Section 3.1.4). We also prepared an explicit aqueduct network by identifying major weirs and manually digitizing their channels (Section 3.1.4).

**Table 3 Geographic data used in this study.**

| Category | Variable | Global | Local |
|---|---|---|---|
| Hydrology | Flow direction | Same as local | Derived from the global elevation data of Becker et al. (2009) |
| | Locations of dams | Same as local | Japan Dam Foundation (Table 2) |
| | Locations of aqueducts (implicit) | Derived from the original H08 algorithm (Hanasaki et al., 2018b) | Derived from the localized algorithm (Section 3.1.4) |
| | Locations of aqueducts (explicit) | Not applicable | Localized data (Section 3.1.4) |
| Agriculture | Distribution of cropland area | Ramankutty et al. (2008) | Kohyama et al. (2003) |
| | Distribution of crop type | Monfreda et al. (2008) | Kohyama et al. (2003) |
| | Distribution of irrigated area | Siebert et al. (2005) | Kohyama et al. (2003) |
| | Crop calendar | Derived from the original H08 algorithm (Hanasaki et al., 2008a) | Localized data (Section 3.2.3) |
| | Irrigation efficiency | Döll and Siebert (2002) | Localized data (Section 3.2.3) |
| | Groundwater fraction of irrigation | Siebert et al. (2010) | Same as global |
| Municipal | Industrial water withdrawal | Food and Agriculture Organization (FAO) (2017) | Ministry of Economy Trade and Industry (METI) (2018) |
| | Domestic water withdrawal | Food and Agriculture Organization (FAO) (2017) | Japan Water Works Association (JWWA) (2018) |
| | Groundwater fraction of industrial water | International Groundwater Resources Assessment Centre (IGRAC) (2004) | Ministry of Land Infrastructure Transportation and Tourism (MLIT) (2021) |
| | Groundwater fraction of domestic water | International Groundwater Resources Assessment Centre (IGRAC) (2004) | Ministry of Land Infrastructure Transportation and Tourism (MLIT) (2021) |

### 3.2.3 Geographic data – Agriculture

We adopted the data of Kohyama et al. (2003) for the distributions of crop land, crop type, and irrigated area, which are crucial factors in this simulation. They developed these data by combining National Land Numerical Information (NLNI) provided by the Ministry of Land Infrastructure and Transportation (MLIT) with the Census of Agriculture and Forestry (CAF; https://www.maff.go.jp/e/data/stat/) by the Ministry of Agriculture, Forestry and Fisheries (MAFF). NLNI is the fundamental grid-based digital land-use map for Japan, which is available at 1-km and 100-m spatial resolutions. One drawback of NLNI is that the cropland area reported in NLNI differs considerably from ground-based agricultural surveys due to issues with land-use classification and spatial resolution (Kohyama et al., 2003). CAF is the most comprehensive agricultural survey in Japan; it is conducted every 5 years, with the latest census conducted in 2015 and published in 2017. CAF is based on interviews with individual farmers and agronomical industries and summarized for each agricultural village, which number approximately 140,000. Although comprehensive and precise, CAF is not available in a grid-based format. Kohyama et al. (2003) scaled NLNI so that the total areas of paddy and upland crops agree with those in CAF for every agricultural village from 1970 to 1995. Unfortunately, for technical reasons, the data developers have stopped updating the dataset since then. Although outdated, we used the gridded data for 1995 or the latest available data in this study. The spatial resolution is 45" × 30" (equivalent to approximately 1 km × 1 km). The number of crop types is 13: rice, barley/wheat, other cereals, root vegetable, green vegetable, legume, flowers, seeds, fodder, orchard, tea, cash crops including sugar cane and sugar beet, and other crops.

The default H08 model subdivides the grid cells into four land-use types, namely double-crop irrigated cropland (for warm regions only), single-crop irrigated cropland, rainfed cropland, and non-cropland (Figure 4a). The primary and secondary crop types are assumed to be cultivated across the entire grid cell (Hanasaki et al., 2008a). In this study, we first subdivided cropland into two categories, paddy and upland, as irrigation water use differs sharply between these two farming methods. Then, we subdivided the grid cells into five land-use types with specific crop rotations, namely double-crop irrigated paddy (rice and barley), single-crop irrigated paddy (rice), single-crop irrigated upland (summer crop), rainfed upland (summer crop), and non-cropland (Figure 4b). The crop parameters of the crop growth sub-model were taken from SWAT (Arnold et al., 2012). We used the "generic" for summer crop.

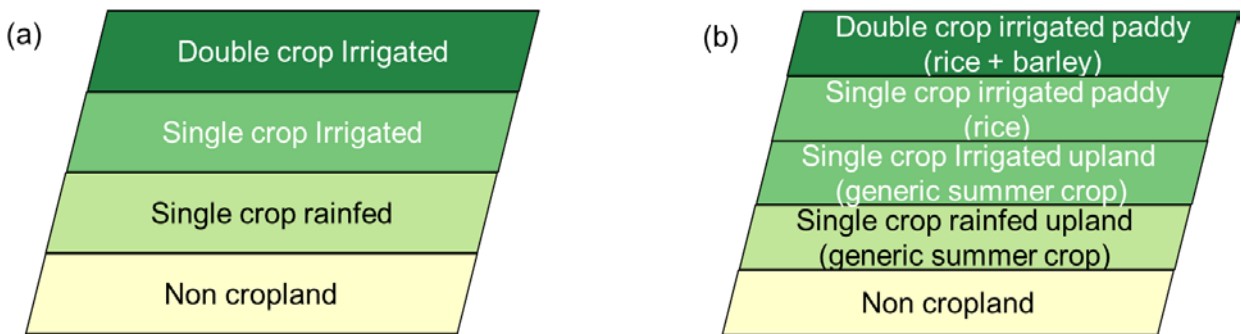

**Figure 4 Grid cell categorization: a) the original categorization for global mode, and b) the new categorization for localized mode.**


To estimate the crop calendars of three crops, we used the stand-alone crop growth sub-model (See Section 3.1.1). In previous global applications of H08, the estimated calendar of major crops agrees well with the reported calendar, which supports the validity of this approach. However, in application to Kyushu, we noted that special treatment is needed for rice. The H08 model estimates the optimal planting date for rice as early May for most areas of Kyushu. However, the actual

cropping practice indicates planting in early June. Therefore, we shifted the planting date of rice by 30 days. For other crops, we used the estimated crop calendar as provided.

Irrigation efficiency is defined as the rate of water consumption (i.e., evapotranspiration) to total water withdrawal. In general, irrigation efficiency is determined by irrigation equipment (i.e., high efficiency for drip irrigation and low efficiency for flood irrigation). Döll and Siebert (2002) adopted this approach and proposed a value of 0.35 for East Asia. Through

comparison between simulation results and local observations (reported in Section 4.1.2), we obtained a value of 0.45 for surface irrigation in this study, which is by far the predominant irrigation type used on Kyushu Island. Lastly, for the fraction of irrigation obtained from groundwater, we were unable to find reliable statistics that cover the entire island. Therefore, we did not prepare local data and used only the global dataset (Siebert et al., 2010).

**3.2.4 Geographic data – Municipal**

Industrial water withdrawal is provided as a census (Ministry of Economy Trade and Industry (METI), 2018). The finest geographical data are provided at the prefecture level, and are further divided into 24 industrial classes. As the prefecture level is too crude (with 7 in the study domain), we downscaled the data to the municipality level (numbering 233) through weighting by the value of exports, which is available for each municipality and industrial class. Finally, gridded industrial

water withdrawal data were developed by assuming that the spatial distribution of water usage is proportional to population (Statistic Bureau of Japan, 2015). All data were prepared for the year 2015 to maintain consistency between municipality-based and grid-based data.

Domestic water withdrawal data were obtained from the Waterworks Statistics (Japan Water Works Association (JWWA),

2018). As the finest resolution available was the prefecture level, gridded data were developed in a manner similar to that of

industrial water. The fraction of groundwater withdrawal for industrial and domestic uses was obtained from the yearbook

(Ministry of Land Infrastructure Transportation and Tourism (MLIT), 2021). These data are available only at the regional

scale (Kyushu Island is subdivided into two regions) for the industrial and domestic sectors (data for the agriculture sector is

not available).

### 3.2.5 Validation data (streamflow, reservoir operation, and irrigation)


To validate the major outputs, we collected the observational data listed in Table 4. We used the Water Information System

(http://www1.river.go.jp/) of the Ministry of Land Infrastructure and Transportation (MLIT) for daily river discharge and

dam operation data (i.e., release, inflow, and storage). In cases where storage is provided only as water level, the value was

converted into volume by fitting a quadratic function to the storage and water level relationship at the total, effective, and

dead storage levels, which are publicly disclosed.

Daily irrigation water withdrawal is not published in Japan. We requested daily irrigation water withdrawal data from the

Kyushu Regional Agricultural Administration Office (KRAAO) of the Ministry of Agriculture, Forestry and Fisheries

(MAFF). KRAAO kindly provided daily irrigation water withdrawal data for 18 weirs in seven national irrigation projects on

Kyushu for 11 years, from 2007 to 2018. The data were provided under following conditions. First, the location of weirs

shall not be disclosed. Second, the raw data shall not be published in any document. Third, the redistribution of data shall not

be permitted for any reason. Due to these conditions, we hereafter refer to the irrigation projects as "A–G".

**Table 4 Validation data. MLIT and MAFF represent the Ministry of Land Infrastructure, and Transportation and the Ministry of Agriculture, Forestry, and Fisheries, respectively.**

| Sub-model | Variables | Source |
|---|---|---|
| Land/River | River discharge | Water Information System, MLIT |
| Reservoir | Reservoir inflow | Water Information System, MLIT |
| | Reservoir release | Water Information System, MLIT |
| | Reservoir storage | Water Information System, MLIT |
| Crop growth/ Water withdrawal | Irrigation water withdrawal | Data request from MAFF |


### 3.3 Simulation and analysis

H08 was set up at 1-arcminute spatial resolution and applied to Kyushu Island (Figure 1). Simulation was conducted for 20 years from 1995 to 2014. Due to the availability of irrigation water withdrawal data, we primarily analyzed the results of the final 8 years, 2007 to 2014. Calibration of hydrological parameters was performed using simulation results for 2001–2006.

We conducted two simulations: a global simulation (hereafter GLB), involving running the H08 model at high spatial resolution using the default model and interpolated standard global data, and the localized simulation (LOC), which uses localized H08 model with localized boundary conditions (Table 5).

We conducted LOC variant simulations to investigate the effects of aqueduct settings. Due to substantial differences in water availability (i.e., streamflow) and water demand (affected by irrigated area and population) among grid cells, inter-grid water

transfer plays a crucial role in balancing these terms. We conducted simulations with different settings for aqueducts and examined the effect on the simulation results. LOC-I1 employs only the original implicit aqueduct algorithm. Aqueducts transfer water to only one downstream grid cell (approximately 2 km). Because traditional weirs and canal systems can transfer water across several to dozens of kilometers, LOC-I5 extends the maximum reach of implicit aqueducts to 5 grid cells (9 km). Water-insufficient grid cells tend to occur in coastal zones, where the catchment area is rather limited but heavily populated. LOC-I5O allows water transfer outside of the basin, as illustrated in Section 3.1.4. Finally, LOC-I5OE

includes explicit aqueducts (i.e., major weirs and municipal water systems, listed in Table S6).

**Table 5 Simulations. Key differences between the GLB and LOC simulations**

|  | GLB | LOC |
|---|---|---|
| Full name | Global | Localized |
| Source of daily meteorological data | The Era-Interim reanalysis (Dee et al., 2011) | Local meteorological observations (AMeDAS) |
| Spatial resolution of daily meteorological data | 45 arcminutes | 10 arcminutes† |
| Geographic data | Global boundary conditions (Table 3) | Local boundary conditions (Table 3) |
| Hydrological parameters | Default | Localized (Section 3.1.2) |
| Model | Default | Localized (Section 3.1.3–4) |

†117 rain gauges for the study domain of 36,750 km$^2$, which is equivalent to approximately one rain gauge per 10×10-

arcminute square.

# 4 Results

## 4.1 Validation

### 4.1.1 Discharge

Simulated daily river discharge at Senoshita on the Chikugo River is shown in Figure 5 (results for other stations are shown
in Figure S2). Monthly river discharge obtained from LOC simulations reproduced historical observations well throughout the simulation period. Aspects of the timing and the magnitude of the largest peaks in July and some minor ones, as well as the magnitude of discharge in dry months, were similar to observations. Daily river discharge also agreed with the observations for both wet and dry years (i.e., the years with minimum and maximum annual rainfall in the validation period), but the simulated peaks tended to be underestimated, especially minor peaks during the dry period (e.g., from September to
May 2012). Monthly river discharge from GLB simulations also agreed with observations, but the daily results show considerable discrepancies. The magnitude of peaks does not match the observed levels, and the duration of high discharge is apparently longer than the observed period (i.e., the shape of curves is flattened) after a prolonged precipitation event in the GLB simulation (not shown).

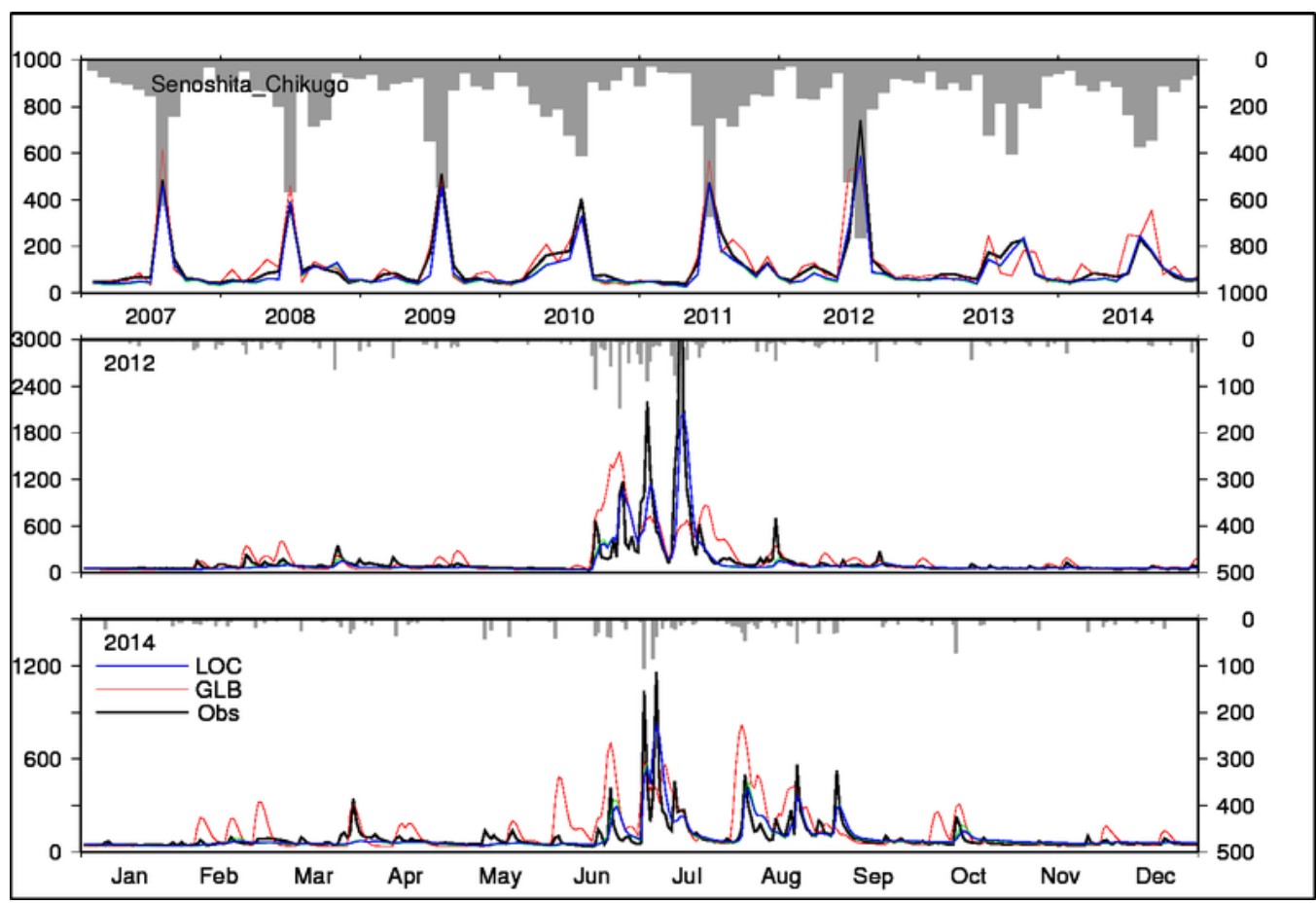


**Figure 5 River discharge at Senoshita on the Chikugo River. Unit is m³ s⁻¹. Top: monthly time series of streamflow (lines) and basin-averaged precipitation in the LOC simulation (bars). Middle: daily time series of streamflow and precipitation in the wettest year of the validation period. Bottom: daily time series of streamflow and precipitation in the driest year of the validation period.**

The results for other basins showed a similar general tendency (Figure S2). The results from LOC for both monthly and daily simulations reproduced the observations. GLB performed moderately well for monthly simulations, but poorly for daily simulations. The NSE values (Nash and Sutcliffe (1970) of all seven rivers are shown in Table 6. The NSE of LOC consistently exceeded 0.6, while all values for GLB were below 0.4.

**Table 6 Nash-Sutcliffe efficiency of daily river discharge simulations.**

| River | GLB | LOC |
|---|---|---|
| Chikugo | 0.33 | 0.68 |

| | | |
|---|---|---|
| Oyodo | 0.35 | 0.65 |
| Kuma | 0.37 | 0.73 |
| Gokase | 0.20 | 0.77 |
| Sendai | 0.22 | 0.82 |
| Ono | 0.05 | 0.63 |
| Midorikawa | 0.02 | 0.73 |
| Onga | 0.36 | 0.71 |
| Kikuchi | 0.20 | 0.71 |

We speculate that the difference in performance between the two simulations is due to the quality of precipitation data. Although the hydrological parameters of LOC were selected from 81 possible combinations, their effect is limited because only one parameter set was applied across the entire study domain (Section 3.1.2). The daily variation in precipitation for GLB is based on the ERA-Interim global reanalysis product with resolution of approximately 45×45 arcminutes (Dee et al., 2011), which is too crude to obtain 1-arcminute spatial data. The variations used for LOC are based on daily precipitation records from 117 rain gauges for the study domain of 36,750 km$^2$, which is equivalent to approximately one rain gauge per 10×10-arcminute square. This high density of observations provided accurate information for the magnitude and timing of each rainfall event. To investigate this speculation a systematic sensitivity simulation for river discharge was conducted (Appendix C).

### 4.1.2 Irrigation water requirement

Figure 6 shows the estimated irrigation water demand at irrigation project A (results for the other six sites are shown in Figure S3). The simulated monthly water requirement of LOC was overestimated by a factor of two, but the duration of irrigation and its temporal variations agreed well with observed water withdrawal. GLB had similar performance to LOC in terms of magnitude but, as discussed above, the irrigation period begins and ends earlier than the observed period by one month.

The daily water requirement results for 2007 and 2010 highlight the differences between the two simulations. First, the daily variations in water requirement are much closer to reality for LOC than for GLB. This difference is due to the meteorological data and hydrological parameters used for each simulation. For example, the smaller τ of LOC (the time constant of interflow; See Supplemental Text Section 1.1) is associated with the overall faster drainage than GLB, which requires a greater volume of daily irrigation to maintain paddy fields without reference to precipitation events. Second, the observed water requirement shows distinct patterns, with low levels in late June and a decrease after September. The former trend reflects precipitation during the rainy season, and the latter tendency represents permanent drainage around 30 days

after flowering. In reality, when rainfall continues during the rainy season, the amount of water withdrawn is greatly restricted, but in H08, water abstraction continues unless there is a considerable amount of rainfall. Note that the rate of decrease is location dependent. For example, irrigation project C (Figure S3) has no clear decrease period, and the simulation agreed rather well with observations at that site.

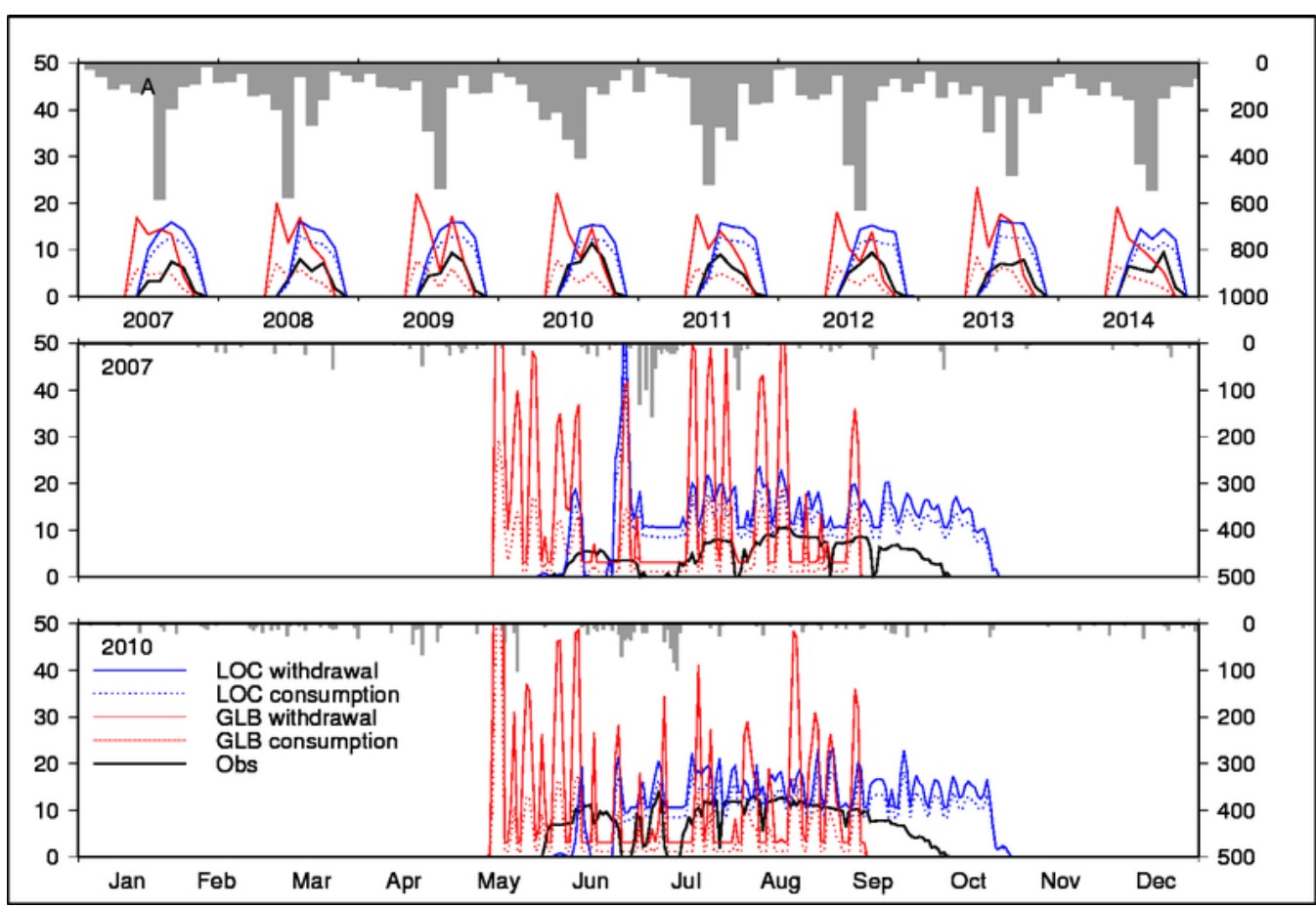

**Figure 6 Irrigation water requirement at irrigation project A. Top: monthly time series of irrigation water withdrawal (lines) and basin-averaged precipitation from the LOC simulation (bars). Solid and broken lines indicate results including and excluding irrigation efficiency, respectively. Middle: daily time series of irrigation water and precipitation in the wettest year during the validation period. Bottom: daily time series of irrigation water and precipitation in the driest year during the validation period.**

Table 7 shows the monthly NSE for the irrigation water requirements of seven irrigation projects. In general, LOC performed well except for projects A (Figure 6) and F. Excluding those two projects, LOC exceeded NSE values of 0.6 and, thus, substantially outperformed GLB, mainly due to the timing of irrigation. The low NSE values in A and F are due to

overestimation of the water requirement. The reasons for such overestimation have not been fully explored, and may be attributed to various possible factors, including the difference between the land-use data period (1995) and simulation period (2007–2014) and region-specific irrigation practices. To investigate the influence of settings, a systematic sensitivity simulation for irrigation requirement was conducted (Appendix D).

**Table 7 Monthly NSE of irrigation requirement.**

| Project | GLB | LOC |
|---|---|---|
| A | -3.24 | -1.39 |
| B | 0.13 | 0.66 |
| C | -0.92 | 0.87 |
| D | -0.56 | 0.81 |
| E | 0.32 | 0.64 |
| F | -16.4 | -9.54 |
| G | 0.17 | 0.87 |

### 4.1.3 Reservoir operation

Figure 7 shows the results for reservoir operation at the Shimouke Dam on the Chikugo River (results for the other five dams are shown in Figure S4). First, the monthly inflow of LOC agrees well with the observations, but shows a tendency toward underestimation of the peaks, which is most apparent in 2009, 2010, 2011, 2012, and 2013. The results of GLB also agree fairly well with observations, but the tendency toward underestimation is more prevalent, including during the dry period. The monthly release estimates of LOC agree moderately well with observations in terms of the arrival of peaks and seasonal variations, while the estimates of GLB do not. In general, GLB tends to overestimate release in winter (October to March, apparent in 2010–2012), leading to frequent depletion (particularly in the winters of 2007, 2009, and 2013). This overestimation is due to the original algorithm of the reservoir operation sub-model of H08, which was designed to release water constantly throughout the year at the rate of mean annual inflow (see 3.1.3). As numerous previous applications have demonstrated, the algorithm works well for the largest reservoirs in the world, in which total storage capacity is greater than or comparable to mean annual inflow (Hanasaki et al., 2006;Masaki et al., 2017;Shin et al., 2019). However, in this reservoir, the storage capacity is only 14% (5–9 % for the remaining reservoirs assessed here) of the mean annual inflow at the dam, and thus is easily depleted during dry periods and overflows in wet periods. The algorithm uses several functions to avoid frequent depletion and overflow events (see Hanasaki et al. (2006)), but they are not applicable to this case because the storage capacity is too small to buffer strong temporal variations in inflow.

LOC and GLB contrast sharply in the performance of their monthly storage simulations. The LOC simulation matched the observed amplitude and temporal variations of storage, while GLB results differed markedly; for example, they included extended periods of depletion (e.g., the winters of 2008, 2010, and 2012). As discussed in the previous paragraphs, the

435 hydrographs for inflow and release generally match well. Such "run-of-the river" operation is often used for relatively small reservoirs located in rivers with high streamflow. To add the maximum possible flow control to the "run-of-the-river" operation, incorporation of the upper and lower storage curves (Figure 2) is beneficial. The seasonal variations of release (Figure 2) also played an important role in stabilizing storage (i.e., not following the upper or lower curve) and release (i.e., not falling below the intended release level) by reducing the gap between inflow and release.

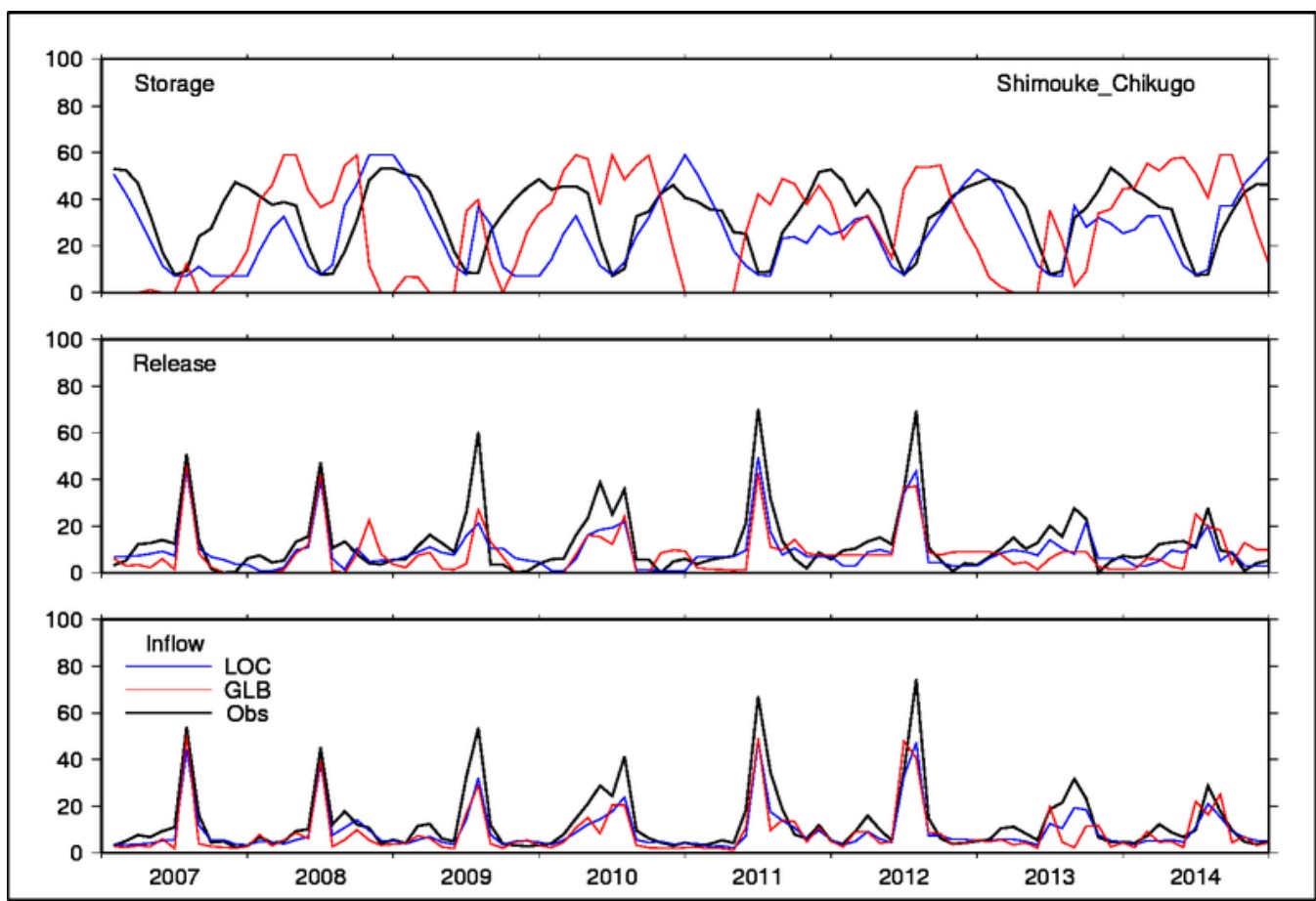

**Figure 7 Reservoir operation at the Shimouke Dam on the Chikugo River. Top: monthly time series of water storage ($10^6$ m$^3$ year$^{-1}$). Middle: monthly time series of release (m$^3$ year$^{-1}$). Bottom: monthly time series of inflow (m$^3$ year$^{-1}$).**

Table 8 shows NSE values for six reservoirs. In general, the performance of LOC in terms of release and inflow is good (i.e.,

NSE > 0.5), except for Iwase Dam. The inflow estimation of GLB is also good, but has a lower NSE than LOC for all dams.

The NSE of release is substantially lower for GLB than for LOC, mainly due to the frequent occurrence of storage depletion, which leads to zero release (Figure 7 and Figure S4). The NSE for storage is negative in all cases, but the values are less negative for LOC than for GLB. The difference in performance, as shown in Figure 7 and Figure S4, is associated with the reproducibility of the temporal variations in storage as discussed in the previous paragraph. Similar to irrigation water

requirement, to investigate the influence of settings, a systematic sensitivity simulation for reservoir operation was conducted (Appendix E).

**Table 8 NSE values for monthly storage, release, and inflow.**

|  | GLB | | | LOC | | |
|---|---|---|---|---|---|---|
|  | Storage | Release | Inflow | Storage | Release | Inflow |
| Shimouke | -3.01 | 0.48 | 0.61 | -0.08 | 0.68 | 0.77 |
| Midorikawa | -52.0 | -0.02 | 0.20 | -7.51 | 0.87 | 0.85 |
| Tsuruda | -12.3 | 0.32 | 0.61 | -3.43 | 0.90 | 0.92 |
| Ichifusa | -11.6 | 0.47 | 0.72 | -0.08 | 0.86 | 0.88 |
| Kitagawa | -3.58 | 0.34 | 0.56 | -0.62 | 0.59 | 0.68 |
| Iwase | -26.7 | -1.20 | -0.80 | -0.95 | -0.77 | -0.85 |

**4.2 Water availability and use balance**

**4.2.1 Water sustainability index**

The results presented thus far indicate that the simulation using localized data and model parameters (LOC) exhibited satisfactory performance for estimation of streamflow in major river basins, irrigation water requirements of major projects, and the operation of major dams. Here, we assess whether water availability and use are balanced. The background for this

assessment is that sufficient water is generally available on Kyushu Island, except for a few locations and events and, therefore, we expect neither a large spatial extent nor a long duration of water scarcity in the study domain. We quantified water balance by calculating the water sustainability index (WSI), as defined in Equation 1.

$$WSI = \frac{\sum(Riv_{DOY} + Aque_{DOY} + RGW_{DOY})}{\sum Req_{DOY}}$$    (1)

where DOY represents day of year and $\sum$ indicates the integration of daily simulation results. Req is the total water

requirement of three sectors. Riv, Aque, RGW are daily water abstraction levels from river, aqueduct, and renewable groundwater sources, respectively. As described in Section 3.1.1, H08 assigns water requirements of the three sectors to five water sources. Among these sources, WSI excludes water abstraction from unspecified surface water and nonrenewable

groundwater, as they indicate overuse and non-sustainable water withdrawal. WSI ranges between 0 (scarcity) and 1 (sufficiency). WSI was first introduced and used by Hanasaki et al. (2008b), and has been applied and modified in numerous studies. Its strengths and limitations relative to other indicators were summarized by Hanasaki et al. (2018a).

The distribution of mean (2007–2014) WSI is shown in Figure 8. The results of LOC-I1 (localized model with implicit aqueducts transferring water only 1 grid cell ahead within the same basin) are shown in Figure 8(a). Under this condition, a substantial number of grid cells indicate low to severe water scarcity, including those very close to major rivers (e.g., the lower Chikugo River east of Saga City). Next, the results of LOC-I5 (same as LOC-I1 but with transfer up to 5 grid cells ahead) are shown in Figure 8(b). Although a substantial number of water insufficient grid cells remained, water scarcity was attenuated across the entire domain because water could be transferred a maximum of five grid cells ahead, connecting grid cells with large water requirements to those with large water availability (e.g., northeast of Fukuoka). Then, the results of LOC-I5O (same as LOC-I5 but allowing water transfer outside of basins with small catchment areas) are shown in Figure 8(c). Because water demand is concentrated along the seashore, inter-basin water transfer reduced water scarcity in coastal areas dramatically (e.g., southwest of Kumamoto). Finally, the results of LOC-I5OE (same as LOC-I5O but including explicit aqueducts) are shown in Figure 8(d). Under this condition, water scarcity was further attenuated (e.g., the lower reach of the Chikugo River near Saga City). Places with exceptionally low WSI are discussed in the next subsection.

The numbers of grid cells with WSI values below 0.99, 0.80 and 0.50 are shown in Table 9. As the total number of grid cells in the study domain is 13,105, approximately 6–8% of grid cells are categorized as experiencing at least low water scarcity (WSI<0.99). Introduction of numerous aqueducts efficiently decreases the number of grid cells with water scarcity, and the effects are stronger for more severe scarcity levels. The number of grid cells is reduced from 1110 (LOC-I1) to 746 (LOC-I5OE) for WSI < 0.99, and from 28 to 1 for WSI < 0.5 with the addition of aqueducts.

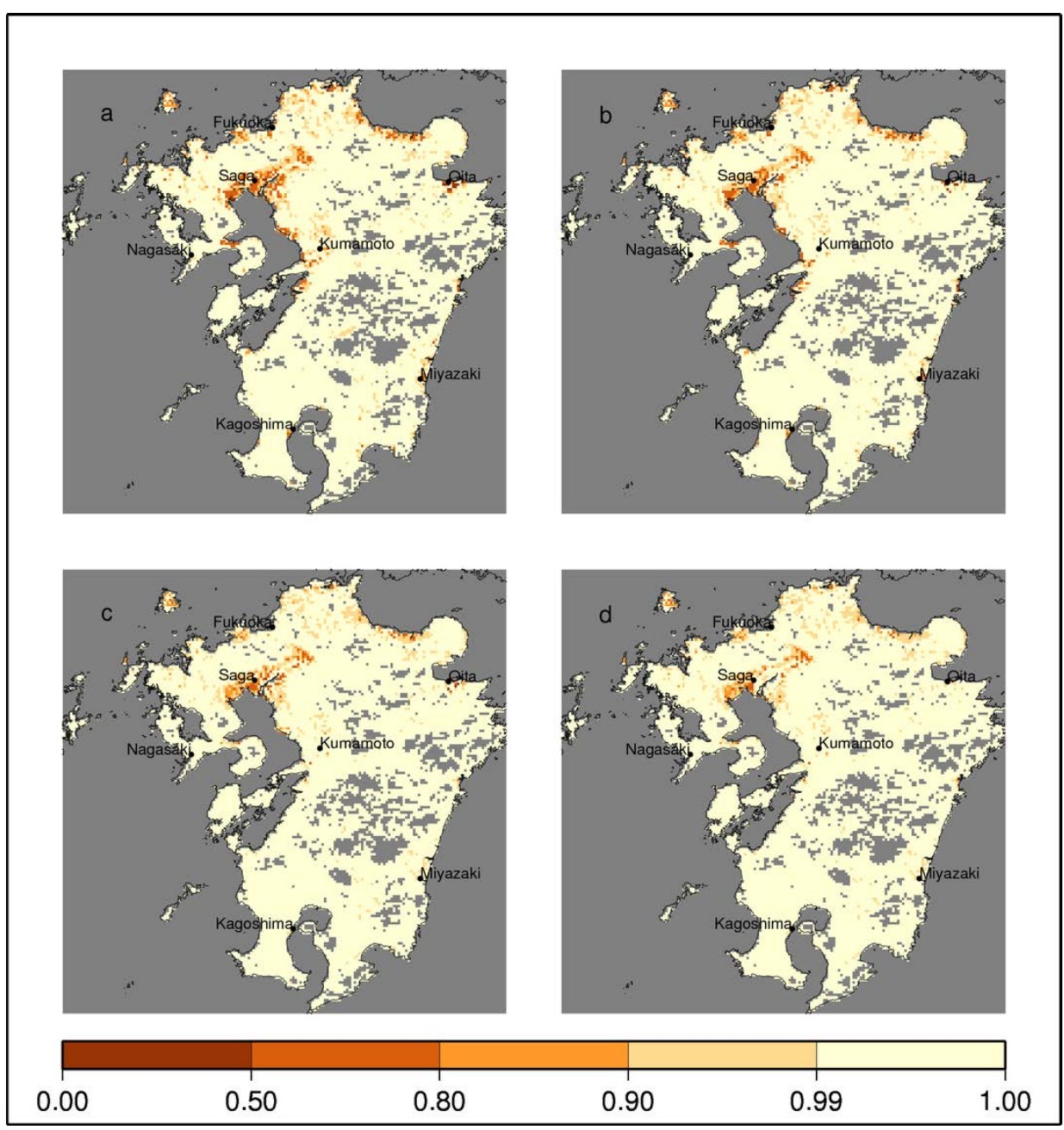

**Figure 8 Water sustainability index. a) LOC-I1 (localized model with implicit aqueducts transferring water to 1 adjacent grid cell), b) LOC-I5 (localized model with implicit aqueducts transferring water to a maximum of five surrounding grid cells), c) LOC-I5O (localized model with implicit aqueducts transferring water to a maximum of five grid cells, including cells outside the basin), d) LOC-I5OE (localized model with implicit aqueducts transferring water to a maximum of five grid cells, including cells outside the basin as well as explicit aqueducts). Symbols show the centers of prefecture capital cities. For visibility, only mean annual withdrawal > 0.03 m³/s is shown.**

**Table 9 Numbers of grid cells below WSI thresholds.**

|          | WSI < 0.5 | WSI < 0.8 | WSI < 0.99 |
|----------|-----------|-----------|------------|
| LOC-I1   | 28        | 253       | 1110       |
| LOC-I5   | 16        | 191       | 941        |
| LOC-I5O  | 5         | 88        | 795        |
| LOC-I5OE | 1         | 54        | 746        |

### 4.2.2 Locations with low WSI

We investigated locations with medium water scarcity (WSI < 0.8) in the LOC-I5OE simulation. Such grid cells are concentrated in four dominant regions, namely the Saga Plain (near Saga City), the Shioishi Plain (southwest of Saga City, connected to the Saga Plain), the Kitano Plain (southeast of Fukuoka City), and Usa City (northwest of Oita City; See Figure S5). We collected information about these regions with the goal of clarifying the reason for low WSI. The information sources used are listed in Table S7, but most are available only in Japanese.

The Saga Plain contains vast areas of irrigated cropland mainly supplied from the Kasegawa River (catchment area 368 km$^2$), which flows near the center of Saga City. The irrigation system in that area consists of the Kitagawa Dam (a single-purpose dam supplying irrigation water to the Saga Plain), the Kawakami Weir, and a dense network of irrigation canals. The LOC-I5OE simulation includes the river, weir, and canals (i.e., explicit aqueducts), but lacks the Kitagawa Dam because its storage capacity of 22.25 10$^6$ m$^3$ is below the threshold of 40 10$^6$ m$^3$ used in this study. Thus, the low WSI estimate is at least partially attributable to underestimation of water availability (i.e., exclusion of the dam limits water supply in the dry period).

Shiroishi Plain is located to the southwest of Saga City, along Ariake Bay. A considerable portion of the plain is reclaimed land that is distant from major rivers. Historically, the area has been irrigated using groundwater pumping. After completion of the Kasegawa Dam in 2012, which is located below the Kitagawa Dam, a new surface water irrigation project was initiated to divert water from the Kasegawa River to this area, but that project was not included in the present study because it was operational only during the last three years of the simulation period (2012–2014). Because no detailed data on the spatial distribution of irrigation water sources were available, we used global data at 5×5-arcminute resolution (Siebert et al., 2010), which indicated that only approximately 7% of the area is appropriate for groundwater irrigation.

The Kitano Plain is located in the southeast of Fukuoka (and northeast of Saga) on the right bank of the middle reach of the Chikugo River. The project in that area is the Ryochiku Heiya Irrigation Project, which is supplied mainly by two tributaries of the Chikugo River, namely the Koishiwaragawa River (catchment area, 85.9 km$^2$) and the Satagawa River (catchment area, 73.6 km$^2$). Each river has one major dam, the Egawa Dam (25.33 10$^6$ m$^3$) and the Terauchi Dam (18 10$^6$ m$^3$), respectively. Similar to the Saga Plain example, these two dams were excluded from this study and, thus, the underestimation is at least partially attributable to the lack of these dams in the simulation.

Usa City is located along the downstream portion of the Yakkan River (catchment area, 430 km$^2$) facing the western extreme of the Seto Inland Sea. Because this area is surrounded by mountains, the basin receives less precipitation than other areas (the mean annual precipitation, 1400 mm year$^{-1}$). Dams on the upstream portion of the Yakkan River and the presence of hundreds of irrigation ponds may play vital roles in the low WSI of this area.

### 4.3 Uncertainty

In this study, we validated the simulated streamflow, irrigation water withdrawal, and reservoir operation for the downstream gauging station of the nine largest rivers, seven national irrigation projects, and six reservoirs on Kyushu Island. The results were reasonably good, indicating that the overall hydrological processes and human water use and management were properly simulated at the scales used for validation. Nonetheless, the water supply and demand balance were not fully closed for all grid cells, indicating that further efforts are needed.

### 4.3.1 Infrastructure

The discussion in the previous subsection indicates that cases of likely water scarcity occur in river basins including reservoirs smaller than the threshold size (i.e., $< 40 \ 10^6$ m$^3$). To incorporate the dams we noted above, the threshold for reservoir storage capacity should be lowered to $18 \ 10^6$ m$^3$ for the study domain. The number of reservoirs meeting this threshold is 29, among which reservoir operation records for 24 are available. Additionally, the major weirs and aqueducts should be georeferenced into the digital river map. Moreover, no severe and extensive droughts were recorded in the study period. During such events, the role of water management infrastructure would be prominent.

### 4.3.2 Groundwater

In this study, we fixed the surface/ground water usage rate to a uniform value across a vast study domain, as the spatial distribution of data was unavailable. As discussed for Shiroishi Plain, some regions intensively use groundwater. Another apparent exception to the uniform value is Kumamoto City, where 100% of municipal water is supplied by groundwater (Kumamoto City, 2015). Aside from usage, the recharge and flow of groundwater may be additional sources of uncertainty. For example, Mount Aso, which is located at the center of Kyushu Island, is an active volcano. The soil in the surrounding area is highly permeable, and forms abundant groundwater. The lack of hydrogeological detail in this model is another source of uncertainty.

### 4.3.3 Municipal water systems

In this study, we have introduced the implicit and explicit aqueducts which allow water withdrawal from surrounding grid cells where water resources are abundant. This is particularly important for urban areas where municipal and industrial water

is highly concentrated. Although improved compared to earlier approaches (e.g. Wada et al. (2016)), which assumed water withdrawal and use occurred in a single grid cell, there is still room for progress. Kyushu Island contains seven cities with populations greater than 400,000, as well as many other settlements. The urban areas of such cities cover dozens of grid cells, which are served by municipal water supply and sewage systems. For urban areas, water is withdrawn at a limited number of inlets to treatment plants and released at outlets from sewage treatment plants. In contrast, H08 allows for water abstraction

and drainage from every grid cell, as it was originally developed for a spatial resolution of 30 arcminutes, at which the effects of urban water systems become negligible. Thus, water scarcity may be underestimated in major cities in this simulation, which allows for usage of water from one grid cell upstream or an intensive cascade of drainage among neighboring cells, but completely ignores the presence of municipal water systems. This problem will be largely solved by including waterwork facilities (i.e. water treatment plants, water and sewer pipe network, and wastewater treatment plants)

into the model. This will make the simulation more realistic in terms of representation of the urban water scarcity problem.

### 4.3.4 Natural hydrological processes

While this study has mainly focused on the processes of water use and management, natural hydrological processes also need to be further improved. Snow falls only in limited mountainous areas of Kyushu Island and, therefore, the performance of snow process estimation was not evaluated in this study. Groundwater is expressed in highly conceptual way in H08

(treated as a hypothetical tank) which hampers the validation of groundwater simulation. As for the lateral flows of subsurface water, Ji et al. (2017) reported that they matter to the distribution of soil moisture and evapotranspiration at the spatial resolution of 1000m and finer. The inclusion of the process is crucial particularly in the regions where groundwater is the major water source. As such, the dominant hydrological processes differ region by region. Further application and investigation of the model under various environment is indispensable to fully realize globally applicable hyper-resolution

modeling.

### 4.4 Lessons for hyper-resolution global hydrological modeling

This study presents a method of validation, localization, and utilization for water resource assessment based on hyper-resolution global hydrological data. The study domain of is 37,500 $km^2$, which is approximately 1/10 of the land area of

Japan (378,000 $km^2$) and 1/3600 of the total land on Earth ($135 \times 10^6$ $km^2$, excluding Antarctica). Eventually, the simulation and validation domains should be extended by three orders of magnitude. In this subsection, we discuss how such global extension may be realized.

### 4.4.1 Validation

River discharge is the easiest factor to validate, but the process has two prerequisites. First, a reliable flow direction map,
which reflects the actual shapes of basins and catchment areas of major river gauging points, is needed. Second, a standardized river discharge publication system is required, and must be well designed for (semi-) automatic data retrieval. The Water Information System used in this study is a good example of such a publication platform, but is optimized for manual data retrieval through mouse operation. Also it is only available in Japanese, hindering data retrieval by non-Japanese-speaking users. Such detailed data provisioning systems are commonly available for developed countries, but not
for other regions. Although efforts have been made to collect historical river discharge records for a considerable number of river basins (van Dijk et al., 2013;Beck et al., 2013), further database development is needed for global hydrological research.

Validation of reservoir operation is more challenging than validation of river discharge. First, reservoir operation records are far less frequently published. This difference is due in part to the fact that reservoirs are operated by multiple agencies. For example, of the 29 reservoirs on Kyushu (i.e., $>18 \ 10^6 \ m^3$), eight are operated by the Ministry of Land Infrastructure
Transport and Tourism (central government), four by the Japan Water Agency (central governmental agency), 13 by prefectures, one by the Ministry of Agriculture Forestry and Fisheries (central government), and three by Kyushu Electric Power Company (private company). Differences in the volume of publicly accessible data exist among these organizations. Satellite-based estimation of reservoir operation is a promising technology (Busker et al., 2019), but may be difficult to apply to Kyushu due to its small lake area and frequent cloud coverage.

### 4.4.2 Data localization

Elevation and river topography data are available globally at 3-arcsecond or 90-m resolution (Yamazaki et al., 2017;Yamazaki et al., 2019). A central data issue is upscaling of such fine-scale data to coarser (e.g., 3-arcsecond to 30-arcsecond) resolution, which is not a trivial problem (Yamazaki et al., 2009). In addition, proper transfer of gauging station and infrastructure (e.g., reservoirs, weirs, canal networks) data is essential. Collection and georeferencing of data for
aqueducts and weirs is a challenging issue. Although previous attempts to estimate the water transfer of large cities (McDonald et al., 2014) and megaprojects (Shumilova et al., 2018) have been reported, far more extensive efforts are needed to achieve global coverage, including small projects (neither of the cited works covered any of the aqueducts included in this study).

Most global agricultural data are available at 5-arcminute resolution (Ramankutty et al., 2008;Monfreda et al., 2008;Siebert
et al., 2010;Siebert et al., 2005). As noted above, 5-arcminute resolution is too coarse for use with 1-arcminute or finer resolution. Because land use and crop type significantly affect water demand, their areal information should be obtained from a national agricultural census. Digitizing and georeferencing census results is crucially important (Kohyama et al., 2003). Some factors, including irrigation water withdrawal, fraction of groundwater use, irrigation efficiency, and the crop calendar, are hard to obtain. Systematic collection of data on such everyday agricultural practices is crucial.

Perhaps the most important data type in terms of simulation performance is meteorological data. The overall performance of this simulation depends on good meteorological data, especially high-quality precipitation data. Most global meteorological data currently available have spatial resolution of 30 arcminutes (Weedon et al., 2014;Cucchi et al., 2020). Recent efforts have made some of meteorological data available at 15-arcminute resolution (Beck et al., 2017;Hersbach et al., 2020;Yatagai et al., 2012), but a gap in spatial resolution remains. A community effort with regard to data standardization and sharing is essential to future progress.

### 4.4.3 Model localization

In this study, the model was localized in four aspects, namely hydrological parameters, irrigation parameters, reservoir operation settings, and implicit canal estimation configuration. First, hydrological parameter optimization can be more systematically applied. Yoshida et al. (2021) demonstrated semi-automatic parameter optimization of H08 based on 777 river gauges globally, which is a promising technique for hyper-resolution global application. For irrigation and reservoir parameter settings, the applicability of the methodology devised in this study to other regions remains unknown. Further modification or improvement will likely be needed for application to other parts of Japan and the world, as hydrometeorological conditions differ significantly among regions. Identification of key formulas and parameters to represent various water-use practices will continue. Rigorous data-basing of large-scale aqueducts and further algorithm validation would make the implicit canal inference process applicable to a broad range of regions. Lastly, the validity of implicit canal estimation was demonstrated only indirectly in this study. The results indicate great improvement of the balance between water supply and demand in the study domain, with realistic results for Kyushu Island, but direct validation against a more detailed database of weirs and canals is required for generalizing the origins and destinations of water diversion networks in Japan.

### 4.4.4 Assessment

High-resolution water assessment will provide novel insights into issues related to water resources. For example, the water scarcity distribution shown in Figure 8 is a reminder of how sensitive water availability is to climatic and geographic conditions. Water scarcity may occur around tributaries of major rivers (e.g., the Kitano Plain), although such areas have been largely overlooked in low-resolution modeling efforts. The present study also highlights the importance of aqueducts and reservoirs, which play crucial roles in balancing water demand and supply. Ignoring water infrastructure may result in misleading water resource assessment results.

## 5. Conclusions

In this study, we applied the global hydrological model H08 to Kyushu Island at a spatial resolution of 1 arcminute, to address three research questions. First, in terms of the reproducibility of the default H08 model and simulation settings, the detailed validation results indicate highly negative values, suggesting that a simple increase in the spatial resolution of global simulation does not lead to reliable results at the local scale. The results highlight the importance of rigorous validation of the most important variables and locations. Second, the process of model localization was demonstrated as promising and effective. Localization of the data and model improve the performance to a satisfactory level (e.g., daily streamflow NSE > 0.5). The prerequisites for localization are good availability, accessibility, and trustworthiness of hydrometeorological data, as well as abundant local water resource information, including historical data. Third, as a benefit of hyper-resolution simulation, we confirmed that the model could support city- or irrigation district-wise assessment. The results highlighted the importance of water management facilities (reservoirs, weirs, and canals) and water use information (groundwater usage and irrigation efficiency). We identified various factors that sustain local water use through examination of a spatially detailed simulation that had been largely overlooked in standard global simulation studies, which are typically at 30-arcminute resolution.

This study contributes to hydrological science in three ways. First, this is the first study to provide an example application of a global water resources model to 1-arcminute spatial resolution. Although the study domain was rather limited (3°×3°), it opens the door to application of the model to hyper-resolution global hydrology. Second, this is the first study to provide a detailed comparison of global and localized simulations using exactly the same source code. We demonstrated that global models can be used as regional hydrological models with reasonable effort. Third, based on our findings, we identified gaps in the existing process and proposed steps toward hyper-resolution global water resources modeling.

For practical climate change adaptation planning and water resource development, local human water use and management are non-negligible problems that can be addressed with sophisticated hyper-resolution global modeling. The detailed information obtained from this study can be utilized to support the recent trend toward identifying environmental impacts of corporate activities in a meaningful manner (e.g., CDP Water Security).

## Appendix A Conversion of sunshine duration into downward shortwave radiation

We converted sunshine duration (hour, $s_d$) into downward shortwave radiation (W m$^{-2}$) using the following equation from Kondo (1994).

$$\frac{s_d}{s_{d0}} = 0.244 + 0.511 \left(\frac{N}{N_0}\right) \tag{A1}$$

Where $s_{d0}$ is solar irradiance at the top of the atmosphere (W m$^{-2}$), N is sunshine duration (hour) and $N_0$ is maximum possible sunshine duration (hour).

**Appendix B Estimation of downward longwave radiation**

We estimated downward longwave radiation (W m$^{-2}$, $L_d\!\downarrow$) using the following equation from Kondo (1994).

$$L_d\!\downarrow = \sigma T^4 \left[1 - \left(1 - \frac{L_{df}\!\downarrow}{\sigma T^4}\right)C\right] \qquad (B1)$$


$$L_{df}\!\downarrow = (0.74 + 0.19 log_{10}w_{TOP} + 0.07(log_{10}w_{TOP})^2)\sigma T^4 \qquad (B2)$$

$$C = 0.826\left(\frac{N}{N_0}\right)^3 - 1.234\left(\frac{N}{N_0}\right)^2 + 1.135\left(\frac{N}{N_0}\right) + 0.298 \qquad (B3)$$

Where $\sigma$ is the Stefan-Boltzmann constant, T is air temperature (K), $L_{df}\!\downarrow$ is clear-sky downward longwave radiation (W m$^{-2}$), and $w_{TOP}$ is effective vapor volume (cm).

**Appendix C Sensitivity simulation on river discharge**

We have decomposed the differences between GLB and LOC simulations into three factors: (A) calibration, (B) local meteorological observation, and (C) high spatial resolution of data. The combination of factors is shown in Table C1.

**Table C1 Sensitivity simulation. The simulation names and the combinations of factors.**

| Simulation | YYY | YYN | YNN | NYY | NYN | NNN |
|---|---|---|---|---|---|---|
| Calibration | Yes | Yes | Yes | No | No | No |
| Local meteorological observation | Yes | Yes | No | Yes | Yes | No |
| High resolution | Yes | No | No | Yes | No | No |

YYY and NNN are identical to the LOC and the GLB simulations respectively. YYY and NYY use the high-resolution daily local meteorological observation (AMeDAS). YNN and NNN use the global meteorological data WFDEI (Weedon et al. 2014). YYN and NYN use the "low-resolution AMeDAS" which was prepared as follows. We aggregated the AMeDAS data for 30 x 30 grid cells in 1 arcminute and converted them into one large grid cell in 30 arcminutes (identical to the data resolution of WFDEI) then linearly interpolated again into 30 x 30 grid cells in 1 685  arcminute. In this way, the only spatial resolution of AMeDAS was decayed. YNY and NNY are not available since they require global meteorological data (reanalysis) as fine as 1-arcminute spatial resolution which do not exist as of today.

The results are shown in Figure C1. In general, the calibration increases the NSE for many cases if other conditions are identical (21 out of 27) but the effect (the change in NSE) is limited. Exceptions are seen, for example, in Stations 1, 6,

8 where NYY (without calibration) performs slightly better than YYY (with calibration). This is considered that due to possible overfitting during the calibration period. Next, high spatial resolution increases the NSE for most of the cases (16 out of 18) but again the effect is limited. Exceptions are seen in Stations 2 and 5 where NYN (low resolution) outperforms NYY (high resolution). Finally, local meteorological data significantly increases the NSE for all the cases (18 out of 18, by median 0.43).

The results of sensitivity test clearly indicate that the usage of local meteorological observation is the key factor in the simulation performance. As mentioned earlier, the timing and magnitude of sub-monthly weather events in the global meteorological data (WFDEI) are based on reanalysis which are not always successful in reproducing the timing and the location where weather events occurred.

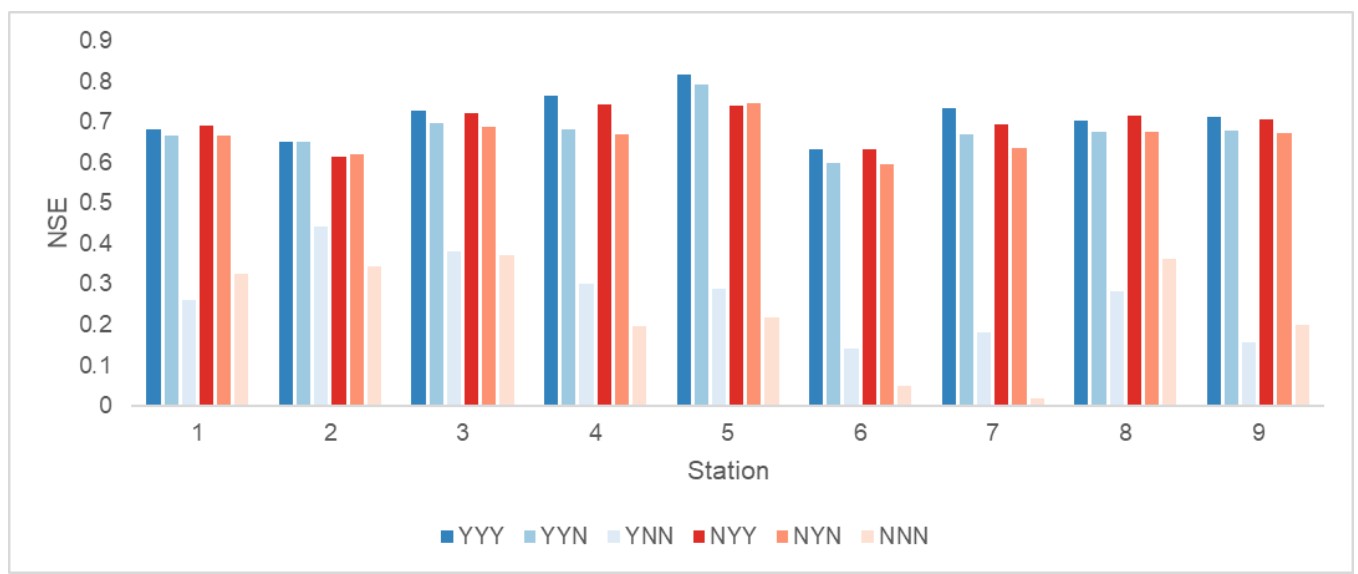


**Figure C1 The results of sensitivity analysis on river discharge. See Table C1 for simulation runs. YYY and NNN correspond to the GLB and LOC simulations, respectively.**

The reproducibility of reanalysis is being improved generation by generation. By using the latest global meteorological data

based on the latest generation of reanalysis, the GLB simulation is expected to perform better (e.g. the WFDE5 (Cucchi et al., 2020) which uses the ERA5 reanalysis (Hersbach et al., 2020)).

**Appendix D Sensitivity simulation on irrigation requirement**

We have decomposed the differences between GLB and LOC simulations into three factors: adoption of (A) local crop calendar, (B) local cropping practices, and (C) local hydrometeorological conditions. The combination of factors is shown in

Table D1.

**Table D1 Sensitivity simulation on irrigation requirement.**

| Simulation | LOC | SI1 | SI2 | SI3 | GLB |
|---|---|---|---|---|---|
| Local crop calendar | Yes | No | No | No | No |
| Local cropping practices | Yes | Yes | No | No | No |
| Local hydrometeorological conditions | Yes | Yes | Yes | No | No |
| Model | LOC | LOC | LOC | LOC | GLB |

The sensitivity simulations SI1, SI2, SI3 were identical to LOC except for the aforementioned three factors. SI1 replaced the adjusted crop calendar for rice with the originally simulated one (Section 3.2.3). SI2 is same as SI1, but additionally replaced the localized settings of land use (Section 3.2.3, Figure 4) and the irrigation efficiency parameter (Section 3.2.3) with those of GLB. Here, crop calendar, land use, and irrigation efficiency are all essential in estimating irrigation water requirement but virtually solely available from governmental reports hence hard to obtain globally. We singled out crop calendar (i.e. separated SI1 and SI2) because it would be available in the future globally since some concrete methods have been proposed by utilizing the latest satellite remote sensing techniques (Kotsuki and Tanaka, 2015). SI3 is same as SI2, but additionally replaced the local meteorological data with the global ones, and the calibrated hydrological parameters with the default ones (Sections 3.1.2 and 3.2.1).

The results are shown in Figure D1. In general, the NSE scores for SI1, SI2, and SI3 are deteriorated in this order, and approaching to those for GLB. The apparent exception is the influence of local cropping practices (i.e. SI2). The irrigation efficiency of GLB is set at 0.35 uniformly while LOC for 0.45-0.8. The adoption of low efficiency inflates irrigation water requirement which drastically decreases the NSE score. The effect is canceled out when the global hydrometeorological conditions were applied which suppresses the total irrigation water requirement (Section 4.1.2).

In contrast to the sensitivity test for discharge shown in Appendix C, the influence of adopting local meteorological condition looks marginal. This is, however, largely due to the considerable drop in the NSE score caused by other factors. This sensitivity simulation highlights the importance of reliable local cropping practices. They can alter the shape, the phase, and the magnitude of simulation timeseries, hence highly influential in the results and performances.

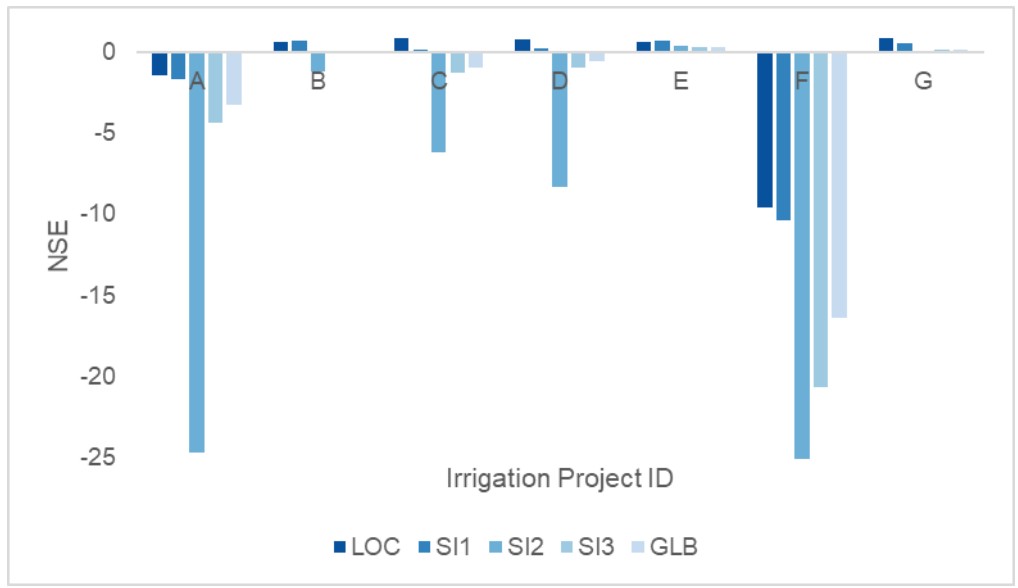

**Figure D1 The results of sensitivity analysis. See Table D1 for simulation runs. The NSE for Irrigation Project F SI2 is -98.4.**


### Appendix E Sensitivity simulation on reservoir operation

We have decomposed the differences between GLB and LOC simulations into three factors: adoption of (A) upper and lower storage rule curves (Figure 2a), (B) release curve (Figure 2b), and (C) local hydrometeorological conditions. The
combination of factors is shown in Table E1.

**Table E1 Sensitivity simulation on reservoir operation.**

| Simulation | LOC | SD1 | SD2 | SD3 | GLB |
|---|---|---|---|---|---|
| Storage curves | Yes | No | No | No | No |
| Release curve | Yes | Yes | No | No | No |
| Local hydrometeorological conditions | Yes | Yes | Yes | No | No |
| Model | LOC | LOC | LOC | LOC | GLB |

The sensitivity simulations SD1, SD2, SD3 were identical to LOC except for the three factors. SD1 removed the setting of
upper and lower storage curves (Section 3.1.3, Figure 2). This setting allows the storage to fluctuate at any time between completely depleted and completely full. SD2 is same as SD1, but additionally removed the release curve. This setting fixes

the release throughout a year at the mean annual inflow (see Hanasaki et al. (2006) for the complete formulations). Although these curves are essential in the actual reservoir operation, it is hard to obtain globally. The advances in satellite global reservoir monitoring would offer the opportunity to estimate the upper and lower storage curves for major reservoirs (Busker
et al., 2019). Hence, we considered that the chance of earning the storage curves (SD1) is more likely than the release one (SD2). SD3 is same as SD2, but additionally replaced the local meteorological data with the global ones, and the calibrated hydrological parameters with the default ones (Sections 3.1.2 and 3.2.1).

The results for storage are shown in Figure E1. In general, the NSE scores for SD1, SD2, and SD3 are deteriorated in this order, and approaching to that for GLB. Note that SD3 and GLB do not completely agree because of other settings between
LOC and GLB (e.g. water withdrawal and diversion). The removal of storage curves has the largest impact on the performance of storage simulation, which can be easily expected. The storage capacity of the dams is relatively small compared to their inflow which typically causes strong storage variations. Once the constraints due to storage rule curves are lost, the simulated storage showed considerable variations by reflecting errors in inflow and release.

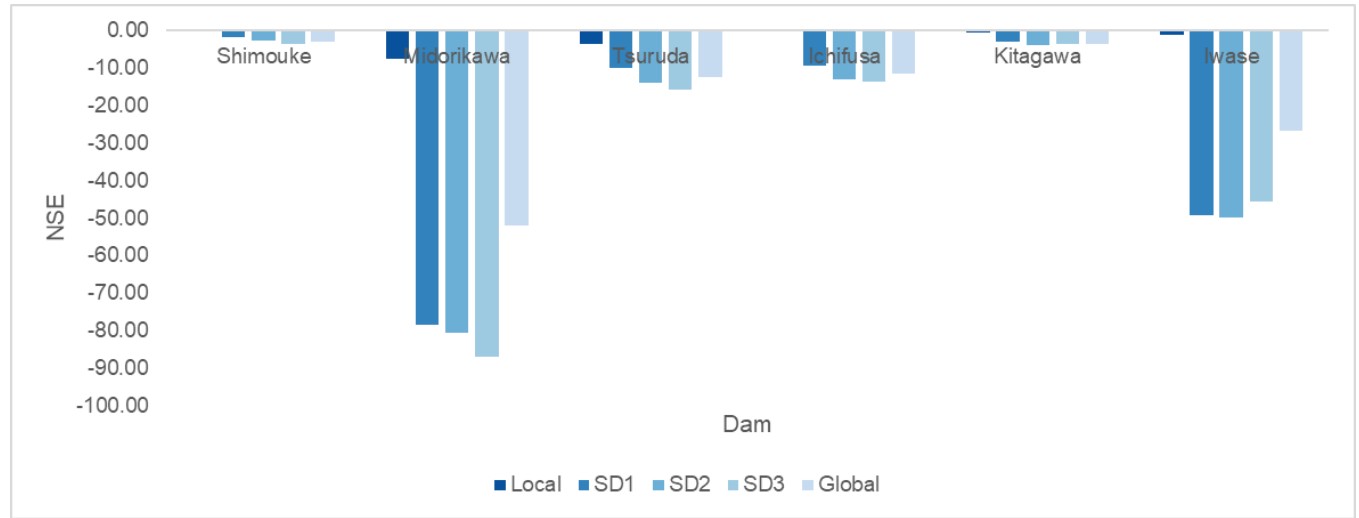


**Figure E1 Sensitivity analysis for reservoir storage. See Table E1 for simulation runs.**

The results for release are shown in Figure E2. Again, the NSE scores for SD1, SD2, and SD3 are deteriorated in this order, and approaching to that for GLB. The impacts due to the adoption of the storage curves, the release curve, and the local
hydrometeorological data are not well discernible except the case for the Midorikawa Dam and the Tsuruda Dam. For these two reservoirs, the adoption of local hydrometeorological data dominantly influenced the simulation performance.

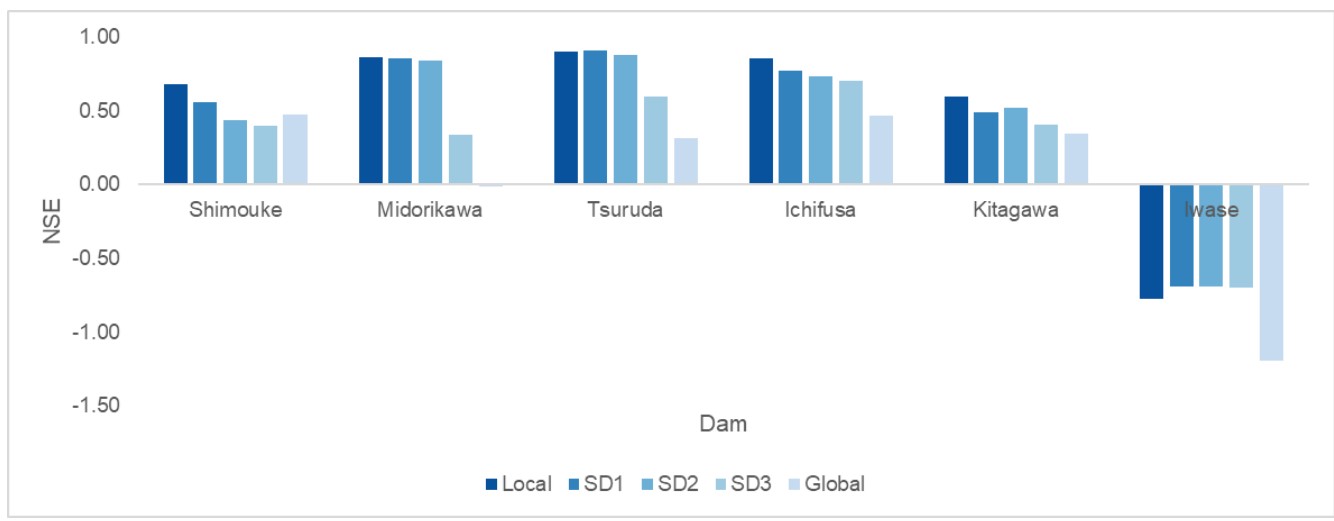

**Figure E2 Sensitivity analysis for reservoir release. See Table E1 for simulation runs.**


Overall, specifying the key factors in the performance of reservoir operation simulation is harder than discharge (Appendix C) and irrigation requirement (Appendix D). This is likely due to the overall low reproducibility of the reality, in particular, the storage variation. Further fundamental improvement is needed in reservoir modeling to reproduce the operations in the past.


**Code and Data Availability**

Code and data are available at https://zenodo.org/record/5523280.

**Author contribution**

NH designed the experiments and NH, HM, and MF carried them out. NH, HM, and MF developed the model code and performed the simulations. YH, SS, SK, and TO validated the model. NH prepared the manuscript with contributions from all co-authors

**Competing interests**

The authors declare that they have no conflict of interest.

## Acknowledgements

The authors are grateful to the three anonymous reviewers and Dr. Luca Brocca for providing comments on this manuscript. This research was supported by KAKENHI (16H06291, 21H05178, and 21H05002) of the Japan Society for the Promotion of Science. The authors are responsible for any errors.

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
