# Peer review of "Toward hyper-resolution global hydrological models including human activities: application to Kyushu Island, Japan"

_Hydrology and Earth System Sciences, 2021_

## Author Comment (AC1)

Dear Dr. Luca Brocca,

Thank you very much for taking the time to review our manuscript. All your comments received were tremendously helpful to improve the paper. We have responded to all the comments below. We believe that all your concerns have been now addressed.

Best regards,
Naota Hanasaki (on behalf of authors)

I read the paper with attention as the topic is highly of interest to me, and I guess to HESS readership. I believe hyper-resolution (1km) global hydrological modelling is an important topic and the number of challenges to be addressed are well underlined in the paper. Indeed, for running hyper-resolution hydrological modelling we need not only High Performance Computing and Storage but also to improve both our modelling capabilities and, likely more important, the observations capabilities at 1km scale, mainly related to the human impact on the water cycle. The paper clearly underline the huge role of water management (agricultural, industrial and domestic), water infrastructures, and the reservoir management at 1km scale and the difficulties to obtain data for that on a global scale.

Thank you for your positive evaluation to our work. We totally agree with your remarks.

I have a short comment related to the results of the hydrological simulation, i.e., river discharge simulations. By reading lines 370-375 it seems that the better performance of localized (LOC) simulation with respect to (wrt) global (GLB) simulation is related the improved spatial resolution of input meteorological data. I believe that improvements are related to: (1) better accuracy of local meteorological data wrt global reanalysis, and (2) LOC model calibration wrt observations. The spatial resolution is likely not as important as written in the paper. It can be easily tested by performing additional experiments.

(A)   Calibrating the GLB model wrt observations as done for LOC model.

(B) Aggregating local observations at larger spatial scale (28x28 arcmin as GLB model input) and check if results will deteriorate (I think deterioration will be only small).

Thank you for this very helpful suggestion. Inspired by your idea, we have carried out a sensitivity simulation and added an appendix as follows.

*Appendix C Sensitivity simulation*

*We have decomposed the differences between GLB and LOC simulations into three factors: (A) Calibration, (B) Local meteorological observation, (C) High spatial resolution of data. The combination of factors is shown in Table A1.*

*Table A1 Sensitivity simulation. The simulation names and the combinations of factors.*

| Simulation | YYY | YYN | YNN | NYY | NYN | NNN |
|---|---|---|---|---|---|---|
| Calibration | Yes | Yes | Yes | No | No | No |
| Local meteorological observation | Yes | Yes | No | Yes | Yes | No |
| High resolution | Yes | No | No | Yes | No | No |

*YYY and NNN are identical to the LOC and the GLB simulations respectively. YYY and NYY use the high-resolution daily local meteorological observation (AMeDAS). YNN and NNN use the global meteorological data WFDEI (Weedon et al. 2014). YYN and NYN use the "low-resolution AMeDAS" which was prepared as follows. We aggregated the AMeDAS data for 30 x 30 grid cells in 1 arc minute and converted them into one large grid cell in 30 arc minutes (identical to the data resolution of WFDEI) then linearly interpolated again into 30 x 30 grid cells in 1 arc minute. In this way, the only spatial resolution of AMeDAS was decayed. YNY and NNY are not available since they require global meteorological data (reanalysis) as fine as 1 arc minute spatial resolution which do not exist as of today.*

*The results are shown in Figure A1. In general, the calibration increases the NSE for many cases if other conditions are identical (21 out of 27) but the effect (the change in NSE) is limited. Exceptions are seen, for example, in Stations 1, 6, 8 where NYY (without calibration) performs slightly better than YYY (with calibration). This is considered that due to possible overfitting during the calibration period. Next, high spatial resolution increases the NSE for most of the cases (16 out of 18) but again the effect is limited. Exceptions are seen in Stations 2 and 5 where NYN (low resolution) outperforms NYY (high resolution). Finally, local meteorological data significantly increases the NSE for all the cases (18 out of 18, by median 0.43).*

*The results of sensitivity test clearly indicate that the usage of local meteorological*

*observation is the key factor in the simulation performance. As mentioned earlier, the timing and magnitude of sub-monthly weather events in the global meteorological data (WFDEI) are based on reanalysis which are not always successful in reproducing the timing and the location where weather events occurred.*

[Figure]

*Figure A1 The results of sensitivity analysis. See Table A1 for simulation runs. YYY and NNN correspond to the GLB and LOC simulations, respectively.*

Minor comment: I believe that, very likely, the use of ERA5 reanalysis (the most recent reanalysis) will improve the results wrt ERA-Interim.

Thank you for this comment. We agree with you that the new generation of reanalysis usually performs better than earlier ones. Very recently, new global meteorological data WFDE5 have been published which use the ERA5 reanalysis. The results of GLB simulation will be better by using the new data, but we speculate they would not fill the aforementioned gap between reanalysis and local meteorological observation. Since it is not practical to redo all the GLB simulations and analyses using WFDE5, let us keep using the present global meteorological data. We have added your point at the end of Appendix as follows.

*The reproducibility of reanalysis is being improved generation by generation. By using the latest global meteorological data based on the latest generation of reanalysis, the GLB simulation is expected to perform better (e.g. the WFDE5 (Cucchi et al. 2020) which uses the ERA5 reanalysis (Hersbach et al. 2020).*

Cucchi, M., Weedon, G. P., Amici, A., Bellouin, N., Lange, S., Müller Schmied, H., Hersbach, H., and Buontempo, C.: WFDE5: bias-adjusted ERA5 reanalysis data for impact studies, Earth Syst. Sci. Data, 12, 2097-2120, 10.5194/essd-12-2097-2020,

2020.

Hersbach, H., Bell, B., Berrisford, P., Hirahara, S., Horányi, A., Muñoz-Sabater, J., Nicolas, J., Peubey, C., Radu, R., Schepers, D., Simmons, A., Soci, C., Abdalla, S., Abellan, X., Balsamo, G., Bechtold, P., Biavati, G., Bidlot, J., Bonavita, M., De Chiara, G., Dahlgren, P., Dee, D., Diamantakis, M., Dragani, R., Flemming, J., Forbes, R., Fuentes, M., Geer, A., Haimberger, L., Healy, S., Hogan, R. J., Hólm, E., Janisková, M., Keeley, S., Laloyaux, P., Lopez, P., Lupu, C., Radnoti, G., de Rosnay, P., Rozum, I., Vamborg, F., Villaume, S., and Thépaut, J.-N.: The ERA5 global reanalysis, Quarterly Journal of the Royal Meteorological Society, 146, 1999-2049, https://doi.org/10.1002/qj.3803, 2020.

---

## Author Comment (AC2)

Dear Reviewer 1,

Thank you very much for taking the time to review our manuscript.

Best regards,
Naota Hanasaki (on behalf of authors)

The paper is very well written.

Thank you for your positive evaluation to this paper.

---

## Author Comment (AC3)

Dear Reviewer 2,

Thank you very much for taking the time to review our manuscript. All your comments received were very helpful to improve the paper. We have responded to all the comments below. We believe that all your concerns have been now addressed.

Best regards,
Naota Hanasaki (on behalf of authors)

Hyper-resolution global water resource modeling is important for provide locally relevant information (e.g., water stress) for the public and policy makers. However, the lack of precise data (e.g., forcing, flow directions with the consideration of facilities, local scale water consumption, local scale dam operation rules and etc.). Hanasaki et al., provide a successful example of high resolution water resource modeling by using the global water resource model. The paper is generally well written, but I still have some major comments before its publication in HESS journal.

Thank you for your positive evaluation to this paper.

Major Comments:

The added value of high resolution meteorological forcing, high resolution hydrological and geographic data (e.g., flow direction, agricultural area, domestic water withdraw) and model improvements (e.g., improved crop classification, new aqueducts scheme) are not clear currently. Although the author discussed this issue in the results and discussion, for example "We speculate that the difference in performance between the two simulations is due to the quality of precipitation data" in L367. I suggest to separate the contributions from these three different factors to make it more clear to the reader that which is the most important for the improvement in high resolution modeling. Sensitive experiments such as "only use high resolution meteorological forcing", "use both high resolution forcing and hydrological data" can be conducted to address the above issue.

Thank you for your suggestion. We agree with you that the improvements in simulation performance should be attributed to individual factors. Taking the advice

by Dr. Luka Brocca (Community Comment), we conducted and analyzed a sensitivity simulation as shown in newly added Appendix C. The results clearly support our claim in the line 367 of the original manuscript. Please see our response to Dr. Luka Brocca for further details.

The inter-grid-cell connection for groundwater. The author give a comprehensive discussion on the challenges of high-resolution modeling and have noticed the importance of inter-grid-cell connection (e.g., the aqueducts and water supply). However, the subsurface or groundwater lateral flow which is important at high resolution is not discussed (Ji et al., 2017). I am wondering whether the H08 model has considered the lateral transport of groundwater from the adjacent grids to the gird which experiences extensive groundwater plumping? If not, whether this process, that is important in hydrology science, lead to uncertainties to the water resource assessment?

Ji, P., Yuan, X., & Liang, X.-Z. (2017). Do Lateral Flows Matter for the Hyperresolution Land Surface Modeling? Journal of Geophysical Research: Atmospheres, 122(22), 12,077-012,092. doi:10.1002/2017jd027366

Thank you for your suggestions. We totally agree with you that lateral groundwater flow is not negligible in hydrological simulations at 1-2 km spatial resolution. From the viewpoint of water resources, it influences the availability of groundwater. Unfortunately, the process is not included in the current H08. To our knowledge, none of the global water resources models (i.e. the global hydrological models including human activities in rich) are able to deal with this flow. Exceptionally, some models offer an option to combine with independent groundwater flow models (e.g. PCR-GLOBWB and MODFLOW; Sutanudjaja et al. 2018, CWatM and MODFLOW; Burek et al. 2020). We have renamed "4.3.4 Other factors" to "4.3.4 Natural hydrological processes" and intensively discuss the uncertainty due to natural hydrological processes. The section now reads as follows.

*While this study has mainly focused on the processes of water use and management, natural hydrological processes also need to be further improved. Snow falls only in limited mountainous areas of Kyushu Island and, therefore, the performance of snow process estimation was not evaluated in this study. Groundwater is expressed in highly conceptual way in H08 (treated as a hypothetical tank) which hampers the validation of groundwater simulation. As for the lateral flows of subsurface water, Ji*

*et al. (2017) reported that they matter to the distribution of soil moisture and evapotranspiration at the spatial resolution of 1000m and finer. The inclusion of the process is crucial particularly in the regions where groundwater is the major water source. As such, the dominant hydrological processes differ region by region. Further application and investigation of the model under various environment is indispensable to fully realize globally applicable hyper-resolution modeling.*

The author review the Wada's work which found "water stress is unrealistically concentrated in grid cells containing the downtown areas of the largest cities (e.g., Paris, New York) and therefore recommended assessment across larger spatial domains, such as sub-basins and counties." While current work wants to assess the water stress at 2 km resolution or even higher resolution in the future. How do you consider or address this unrealistic phenomenon shown by Wada et al., (2016)? Moreover, the author pointed out that "Thus, water scarcity may be underestimated in major cities in this simulation..." this seems to be contrary to Wada's work. Detailed discussions are needed.

Thanks for these important questions. As for the first question, we have resolved the problem by introducing the concept of inter-basin water transfer. The key problem of the approach of Wada et al. (2016) is that the water withdrawal and water use occurred in the single grid cell. Now this study introduced the implicit and explicit aqueducts which allow water withdrawal from surrounding grid cells where water resources are abundant. To make our point clearer, we have added the following text at the beginning of "4.3.3 Municipal water systems".

*In this study, we have introduced the implicit and explicit aqueducts which allow water withdrawal from surrounding grid cells where water resources are abundant. This is particularly important for urban areas where municipal and industrial water is highly concentrated. Although improved compared to earlier approaches (e.g. Wada et al. 2016), which assumed water withdrawal and use occurred in a single grid cell, there is still room for progress.*

Our direct answer to the second question is that we should eventually include anthropogenic water network (i.e. water treatment plants, water pipe and sewer pipe network, and wastewater treatment plants) into the model. Since our intension was unclear in the original text, we have added the following sentence at the very last part of "4.3.3 Municipal water system".

*This problem will be largely solved by including waterwork facilities (i.e. water treatment plants, water and sewer pipe network, and wastewater treatment plants) into the model. This will make the simulation more realistic in terms of representation of the urban water scarcity problem.*

On the difference of using global and local water resource model. The author said that data localization and model localization are important for global high resolution hydrological modeling. As I am not expert at water resource modeling, I am confused that what the difference between high resolution global modeling and high resolution regional modeling? If the difference is due to the input data (e.g., forcing, reservoir operation, water use data), then what the difference between the following pathways: 1. use global water resource model, 2. use regional models (e.g., SWAT) and apply them at all global catchment.?

Thank you for this interesting question. We speculate that eventually the results of a high-resolution global model and those of a regional applied to all basins in the world will become indistinguishable. We also suppose, however, the difference in procedure and concept between two models could remain long. The global models (including H08) are designed to apply globally even if input data were insufficient or details of processes were unknown. Hence the simulations tend to be model-oriented, relying on universal physical or empirical (statistical) relationship to estimate various fields uniformly. On the contrary the regional models (e.g. SWAT) are designed to apply the basins where fundamental input/validation/ boundary condition data are available. Hence the simulations tend to be data-oriented, utilizing parameter estimation techniques.

---

## Author Comment (AC4)

Dear Reviewer 3,

Thank you very much for taking the time to review our manuscript. All your comments received were very helpful to improve the paper. We have responded to all the comments below. We believe that all your concerns have been now addressed.

Best regards,
Naota Hanasaki (on behalf of authors)

The authors present a study comparing the impact of global datasets vs. localized datasets when parameterizing hyper-resolution models for water resources applications. The paper is well written, and I appreciated the authors' efforts to obtain the best possible detailed localized data for their study site. Results show that, as expected, localized models performed better. While this is an interesting exercise, the paper would benefit from a better contextualization and insights on how we can leverage their work/their findings towards hyper-resolution modeling at the global scales, as the title suggests. Here are some major comments that should be addressed before publication:

Thank you for your positive evaluation to this paper.

L 131. "it was designed for applications to any spatial domain and resolution." One of my biggest concerns when applying large-scale designed hydrologic models to hyper-resolutions is the lack of or inappropriate processes representation at the fine scales. These large-scale models were often designed under the large grid cell assumption, in which local-scale and non-linear integrations between water and land and climate were negligible. However, when moving to fine scales, processes such as lateral water flow, the interaction between the river network and the land, and surface water pumping uphill, are no longer negligible. Rather than just increasing computational grid and parameter regionalization, hyper-resolution modeling also comes with the need to further understand and represent hydrologic processes at these scales. The authors do attempt to address the need for improving human activities representation at these scales with the implicit aqueduct estimation. It would be good to have a subsection where the authors describe the H08 efforts towards also improving hydrologic processes representation at the fine scales.

Thank you for this thoughtful comment. We agree with your point that hyper-resolution model require representation of region-specific hydrological processes not only on human activities but also on natural hydrological processes. Taking the comment of you and Reviewer 2, we have modified "4.3.4 Other factors" into "4.3.4 Natural hydrological processes" which now reads as follows.

*While this study has mainly focused on the processes of water use and management, natural hydrological processes also need to be further improved. Snow falls only in limited mountainous areas of Kyushu Island and, therefore, the performance of snow process estimation was not evaluated in this study. Groundwater is expressed in highly conceptual way in H08 (treated as a hypothetical tank) which hampers the validation of groundwater simulation. As for the lateral flows of subsurface water, Ji et al. (2017) reported that they matter to the distribution of soil moisture and evapotranspiration at the spatial resolution of 1000m and finer. The inclusion of the process is crucial particularly in the regions where groundwater is the major water source. As such, the dominant hydrological processes differ region by region. Further application and investigation of the model under various environment is indispensable to fully realize globally applicable hyper-resolution modeling.*

This modeling exercise demonstrated how models with localized inputs perform better than with global inputs. It would be great if the authors could provide a sensitivity analysis of the different input datasets to identify which are the most critical for improved performance. I propose a validation analysis using a leave one input out approach. In this way, besides just reporting what we know is already expected (localized models perform better), this paper has the potential to actually inform the scientific community of which of the inputs for hyper-resolution modeling we should be focusing on improving. Of course, all of them are important, but ranking them would greatly value future work in this field. Is that crop data? Precipitation? Water use and withdraws, etc.

Thank you for this comment. For hydrological simulation, we have newly conducted a sensitivity test. Please see our response to Dr. Luka Brocca for the results. In short, the results indicate that the usage of local meteorological observation dominantly contributed to improving the performance. Similar sensitivity simulations can be done for other components, including irrigation water requirement estimation and dam operation, but we omitted them because we can easily expect earning the same conclusions.

L 630. "it opens the door to applications of the model to hyper-resolution global hydrology." Not really, the results actually show the opposite: that without localized data global hyper-resolution modeling of water resources is inaccurate, and that perhaps we should be focusing on localized models. If the authors can demonstrate what are the large sources of uncertainties (e.g., which are the localized input variables that drive the uncertainties), then you could say that improving a given X input at global scales can enable more accurate hyper-resolution global hydrologic modeling.

> Thank you for your comment. We have now conducted sensitivity analysis and demonstrated that the key factor of improving the performance of hydrological simulation is the usage of daily local meteorological observation. This can be realized in two ways. One is to extensively collect daily local meteorological observation globally perhaps through international cooperation and latest information technologies. The other is to improve the reproducibility of reanalysis in particular the timing and extent of daily weather events (i.e. most of the sub-monthly scale global meteorological data are relying on reanalysis). These points and rationale are concisely summarized in newly added Appendix C (please see also our response to Dr. Luka Brocca).

Model calibration is an important aspect of the model validation performance, as such, the details should be moved from the supplemental material to the main text.

> Thank you. While we agree that model calibration played an important role in this study, we wish to keep placing the part in Supplemental Material because the method itself is very simple and the description of the related models overlaps with earlier papers (e.g. Hanasaki et al. 2008; 2018).

Section 3.1.4, in the implicit aqueduct scheme, does the water only moves downhill? How about water pumping schemes? While they are often negligible at the coarse scale they are very much relevant at the local scale.

> Thank you for this question. First, as shown in text (Line 205 of the original version), we assume the elevation of implicit aqueduct destinations (or the route of implicit aqueducts) must be always lower than the origins. This is an important assumption in the algorithm of implicit aqueduct estimation to express local water management

which sometimes dates back centuries ago when the gravity drainage is virtually the only option. We agree with you that pumps are implemented even if the aqueducts are basically drained by gravity, but these are not resolved in 1 arc minute of spatial resolution. Please note that explicit aqueducts transfer water irrelevant to elevation. Hence the aqueducts on the premise of considerable pumping can be expressed as explicit aqueducts. We have modified the following parts of "3.1.4 Implicit aqueduct estimation".

*… the origins and destinations of known major water transfer facilities were prepared and implemented into the river network as explicit aqueducts. These explicit aqueducts transfer water irrelevant to elevation.*

Moderate/Minor:

Section 4.3, as mentioned earlier, authors should also address hydrologic uncertainties in this section.

Thank you. We have now Section 4.3.4 "Natural hydrological processes" to discuss the hydrologic uncertainties.

Figure 3. This figure could be improved for clarity. Inter-basin transfer you mean between grid cells? Does the grid cell comprise a basin? Are the rives the blue arrows or the blue grids?

Thank you. Now we have modified the schematic for clarity (i.e. the original figure doesn't look like basins). The inter-basin transfer means water in one basin is transferred to another. The gird cells and flow directions comprise basins. As shown in the legend, the blue arrows are the rivers.

[Figure]

To make the point of our modeling clearer, we also modified the caption as follows.

*Figure 3 Implicit aqueducts. a) The original scheme, which does not allow inter-basin transfer. Many coastal grid cells (shown in dark green) are prone to water scarcity because they have limited catchment area. b) The new scheme, which conditionally allows inter-basin transfer. Coastal grid cells receive inter basin water transfer from neighboring basins. Columns represent land grid cells. The height indicates each grid cells' mean elevation. Colors indicate different basins.*

Table 5 would benefit from a more descriptive header.

Thank you. Now the caption reads "Table 5 Simulations. Key differences between the GLB and LOC simulations". To make the table consistent with the sensitivity simulation, the table has been slightly modified. That is, the original "meteorological data" are now subdivided into "spatial resolution of daily meteorological data" and "source of daily meteorological data".

---

## Referee Report (RR1)

The authors have clarified most of the comments and concerns. However, one major concern remains: authors should include and analysis the uncertainties and sensitivity of the water management components. While large-scale hydrologic models have been developed and validated under the uncertainties and assumptions that holds at the large-scale (e.g. 1 deg resolution), when moving to hyper-res processes should be revised (including water management) and uncertainties should be understood. Since the focus of this paper is the implementation of water management at hyper-res scale and potentials to expand it to global scales, understanding of the uncertainties (and sensitivity) of the water management components at the hyper-res scale must be address prior to publication.

**Previous comments and replies:**

**Review comment:** This modeling exercise demonstrated how models with localized inputs perform better than with global inputs. It would be great if the authors could provide a sensitivity analysis of the different input datasets to identify which are the most critical for improved performance. I propose a validation analysis using a leave one input out approach. In this way, besides just reporting what we know is already expected (localized models perform better), this paper has the potential to actually inform the scientific community of which of the inputs for hyper-resolution modeling we should be focusing on improving. Of course, all of them are important, but ranking them would greatly value future work in this field. Is that crop data? Precipitation? Water use and withdraws, etc.

*Author Reply: Thank you for this comment. For hydrological simulation, we have newly conducted a sensitivity test. Please see our response to Dr. Luka Brocca for the results. In short, the results indicate that the usage of local meteorological observation dominantly contributed to improving the performance. Similar sensitivity simulations can be done for other components, including irrigation water requirement estimation and dam operation, but we omitted them because we can easily expect earning the same conclusions.*

Review Reply: I appreciate the authors efforts to include the sensitivity analysis on the precipitation products, as suggested by the other reviewer. I suggest moving it to the main body of the manuscript. However, the main objective of this paper is the implementation and assessment of water management at hyper-res scales and the potentials of expanding it to global scales, isn't it? The authors should include the sensitivity analysis of the water management components (irrigation and dam operation are great examples). Although the authors expect the same conclusions, in my opinion, the uncertainties in water management components or human influence processes are far much larger and must be understood before applying and using these models to simulate water management at the local scales. Quantify these uncertainties will provide confidence (or not) that such processes are well represented at H08 at the hyper-res scale. Furthermore, it can provide the scientific community with novel insights on what needs to be done or pathways forward to implement such water management components at global scales. Many papers have already quantified the uncertainties of precipitation data quality to hyper-res modeling, this manuscript has the unique opportunity to be the first to quantify the uncertainties of the water management component at this scale.

---

## Author Response (AR2)

Dear Editor and Reviewer,

Thank you for taking the time to review our paper again. We fully understand your point and have conducted a sensitivity study for the water management components. We hope that the revision shown below meets your expectations.

Best wishes,
Naota Hanasaki (on behalf of authors)

*The authors have clarified most of the comments and concerns. However, one major concern remains: authors should include and analysis the uncertainties and sensitivity of the water management components. While large-scale hydrologic models have been developed and validated under the uncertainties and assumptions that holds at the large-scale (e.g. 1 deg resolution), when moving to hyper-res processes should be revised (including water management) and uncertainties should be understood. Since the focus of this paper is the implementation of water management at hyper-res scale and potentials to expand it to global scales, understanding of the uncertainties (and sensitivity) of the water management components at the hyper-res scale must be address prior to publication.*

> Thank you for your insightful words. Our response is shown below in detail.
> During the revision, we noticed that the script for making figures and tables for irrigation requirement (Figure 6, Figure S3, and Table 7) contained an error. That is the irrigation efficiency for the GLB simulation was identical to LOC by mistake. We have corrected the figures, the table, and relevant text. Please be informed that the conclusions are not affected by this correction.

*Previous comments and replies:*
*Review comment: This modeling exercise demonstrated how models with localized inputs perform better than with global inputs. It would be great if the authors could provide a sensitivity analysis of the different input datasets to identify which are the most critical for improved performance. I propose a validation analysis using a leave one input out approach. In this way, besides just reporting what we know is already expected (localized models perform better), this paper has the potential to actually inform the scientific community of which of the inputs for hyperresolution modeling we should be focusing on improving. Of course, all*

*of them are important, but ranking them would greatly value future work in this field. Is that crop data? Precipitation? Water use and withdraws, etc.*

*Author Reply: Thank you for this comment. For hydrological simulation, we have newly conducted a sensitivity test. Please see our response to Dr. Luka Brocca for the results. In short, the results indicate that the usage of local meteorological observation dominantly contributed to improving the performance. Similar sensitivity simulations can be done for other components, including irrigation water requirement estimation and dam operation, but we omitted them because we can easily expect earning the same conclusions.*

*Review Reply: I appreciate the authors efforts to include the sensitivity analysis on the precipitation products, as suggested by the other reviewer. I suggest moving it to the main body of the manuscript. However, the main objective of this paper is the implementation and assessment of water management at hyper-res scales and the potentials of expanding it to global scales, isn't it? The authors should include the sensitivity analysis of the water management components (irrigation and dam operation are great examples). Although the authors expect the same conclusions, in my opinion, the uncertainties in water management components or human influence processes are far much larger and must be understood before applying and using these models to simulate water management at the local scales. Quantify these uncertainties will provide confidence (or not) that such processes are well represented at H08 at the hyper-res scale. Furthermore, it can provide the scientific community with novel insights on what needs to be done or pathways forward to implement such water management components at global scales. Many papers have already quantified the uncertainties of precipitation data quality to hyper-res modeling, this manuscript has the unique opportunity*

Thank you for this comment. We have noticed the necessity of conducting some sensitivity tests for this study before the initial submission, but we have not implemented them because of the difficulty in designing such simulations. The difference between the localized (LOC) and the default (GLB) simulations is numerous, hence the number of combination of options/parameters can be easily exploded. Based on your advice, we have returned to a deep consideration what the most informative and efficient way of sensitivity analysis could be. Partly based on the previously added sensitivity test for natural hydrological components (i.e. Appendix C), we have newly added two elaborated sensitivity tests, one for irrigation water requirement and the other for dam operation. Because the structure of model is substantially different between the localized and the default global models hence

the results are not always easy to interpret. Nevertheless, the discussion should be insightful for the scientific community, we believe. Last but not least, please accept the current style that is to place independent appendices for sensitivity analysis. We have tried to incorporate the sensitivity analyses in body for many times, but finally we abandoned it because it requires placing the descriptions of simulations and results in multiple sections (i.e. at least in the methods and the results sections, respectively) which makes the manuscript long, complex, and unreadable.

[revised manuscript text omitted]